# Intracranial electrophysiological and structural basis of BOLD functional connectivity in human brain white matter

Yali Huang [1,6], Peng-Hu Wei [2,6], Longzhou Xu[1,3,6], Desheng Chen[2], Yanfeng Yang[2], Wenkai Song[1], Yangyang Yi[1], Xiaoli Jia[1], Guowei Wu[1], Qingchen Fan[1], Zaixu Cui [1] ✉ & Guoguang Zhao [2,4,5] ✉

While functional MRI (fMRI) studies have mainly focused on gray matter, recent studies have consistently found that blood-oxygenation-level-dependent (BOLD) signals can be reliably detected in white matter, and functional connectivity (FC) has been organized into distributed networks in white matter. Nevertheless, it remains unclear whether this white matter FC reflects underlying electrophysiological synchronization. To address this question, we employ intracranial stereotactic-electroencephalography (SEEG) and resting-state fMRI data from a group of 16 patients with drug-resistant epilepsy. We find that BOLD FC is correlated with SEEG FC in white matter, and this result is consistent across a wide range of frequency bands for each participant. By including diffusion spectrum imaging data, we also find that white matter FC from both SEEG and fMRI are correlated with white matter structural connectivity, suggesting that anatomical fiber tracts underlie the functional synchronization in white matter. These results provide evidence for the electrophysiological and structural basis of white matter BOLD FC, which could be a potential biomarker for psychiatric and neurological disorders.

Functional MRI (fMRI) has been extensively used to localize neural activity based on the blood-oxygenation-level-dependent (BOLD) contrast in the human brain[1–3]. While fMRI studies have mainly focused on gray matter, recent evidence from multiple independent efforts has demonstrated that BOLD signals can be reliably detected in white matter in the resting state and various task states[4–13]. These studies have consistently shown that white matter BOLD signals are not noise, as previously thought, but rather exhibit distinct patterns with both tract- and task-specific power spectra, which could be related to the underlying neural activity[4–6,8,11,12,14]. Moreover, recent studies have characterized the hemodynamic response function[5,10,15,16] and neuroplasticity[17–19] in white matter using BOLD fMRI.

By measuring the temporal synchronization of resting-state BOLD signals—termed 'functional connectivity (FC)'—between two white matter regions, prior studies found that white matter displayed an intrinsic organization of interacting functional networks[7,20–27], similar to those in gray matter. For example, Peer et al. parcellated the white matter into 12 symmetrical functional networks, which were organized into three layers with distinct levels of correlation with cortical gray matter functional networks[20]. Huang et al. further demonstrated that the white matter functional networks were highly reproducible across two independent datasets, and that these networks were organized into two groups with anti-correlated connectivity[21]. Moreover, it has been shown that white matter BOLD FC is constrained by the

[1]Chinese Institute for Brain Research, Beijing 102206, China. [2]Department of Neurosurgery, Xuanwu Hospital Capital Medical University, Beijing 100053, China. [3]State Key Laboratory of Cognitive Neuroscience and Learning, Beijing Normal University, Beijing 100875, China. [4]National Medical Center for Neurological Diseases, Beijing 100053, China. [5]Beijing Municipal Geriatric Medical Research Center, Beijing 100053, China. [6]These authors contributed equally: Yali Huang, Peng-Hu Wei, Longzhou Xu. ✉e-mail: cuizaixu@cibr.ac.cn; ggzhao@vip.sina.com

structure of anatomical white matter tracts[20,21] and is encoded in gene expression profiles[25]. Prior studies also suggest that white matter BOLD FC could be a neuromarker for multiple psychiatric and neurological disorders, including schizophrenia[28,29], depression[30], Alzheimer's disease[31], and Parkinson's disease[32,33]. However, it remains unclear whether the white matter BOLD FC reflects underlying neural synchronization of intracranial electrophysiological signals in the white matter or merely a vascular phenomenon.

Intracranial EEG (IEEG) is an invasive approach for recording local field potentials (LFPs) in the brain to identify the precise origin of seizures in drug-resistant epilepsy[34]. It typically includes electrocorticography (ECoG), which comprises implanted electrode grids on the exposed cortical surface, and stereotactic EEG (SEEG), which comprises depth electrodes penetrating the brain[34]. In a landmark study, Betzel et al. demonstrated that intracranial electrophysiological FC, defined as the correlation between LFP time series from two ECoG electrodes, shared a similar network structure with BOLD FC in the gray matter[35]. However, it is unknown if this is also the case for white matter. In contrast to ECoG, SEEG electrodes typically penetrate the brain through white matter and have 4–18 contacts with a center-to-center space between two adjacent contacts ranging from 2–10 mm[36], providing an opportunity to record LFPs in white matter tissues. Recently, Revell et al. revealed that white matter FC is stronger than gray matter FC, although white matter signals were weaker in SEEG data[37]. However, this study did not seek correlations between electrophysiological signals with BOLD FC in white matter.

In this study, we aimed to provide evidence for an intracranial electrophysiological basis of white matter BOLD FC using SEEG data. Since prior studies have consistently demonstrated that white matter connectivity serves as a structural basis for the functional communication dynamics between brain regions[35,38], we hypothesized that both BOLD and SEEG white matter FC are constrained by the underlying white matter structural connectivity. We tested these predictions using a multimodal dataset from a group of 16 patients with drug-resistant epilepsy, with each one completed intracranial SEEG, non-invasive resting-state BOLD fMRI, and high-quality diffusion spectrum imaging (DSI, ~24 min acquisition). Our results indicated that BOLD white matter FC was highly correlated with SEEG white matter FC across a wide range of frequency bands in every participant, and both BOLD and SEEG FC were highly correlated with structural connectivity in the white matter.

## Results

### BOLD and SEEG white matter FC

We studied intracranial SEEG recordings during the interictal period in 16 patients with drug-resistant epilepsy who needed SEEG to localize seizure onset (Table S1). Each participant had 6–12 electrodes and each electrode had 5–16 contacts. Using an a priori White Matter Parcellation Map (WMPM) atlas[39,40], we found that white matter contacts were mainly localized in the temporal, frontal, and parieto-temporal areas (Fig. S1a and Fig. S1b, See Supplementary Information for details). All participants underwent preoperative structural MRI, fMRI, DSI, and X-ray computed tomography (CT). We localized the coordinates of all contacts based on structural MRI and CT data and then identified the contacts within the white matter. SEEG data were processed and filtered into seven frequency bands (1–4, 4–8, 8–13, 13–30, 30–40, 40–70, and 70–170 Hz) as in prior work[41]. We calculated Pearson's correlations between the time series from all white matter contacts to generate the SEEG white matter FC. Next, we estimated the BOLD white matter FC by computing Pearson's correlations between BOLD fMRI time series from all ROIs, which were defined as spheres in which the white matter contacts were centered.

### BOLD and SEEG white matter FC are highly correlated in a single participant

We first evaluated the correspondence between BOLD and SEEG white matter FC in a single participant (sub1, see participant information in Table S1). To visualize the white matter FC, we displayed the time series of both SEEG (1–4 Hz) and BOLD signals at two white matter contacts (Fig. 1a) with the MNI coordinates (46, −14, −20) and (34, −4, −23), respectively. Both the SEEG and BOLD time series were highly synchronized by visual inspection, suggesting FC between the two white matter contacts in both BOLD fMRI ($r = 0.52$) and SEEG ($r = 0.55$) data. Next, we depicted the upper triangle of BOLD and SEEG (1–4 Hz) white matter FC matrices side-by-side, which presented a highly similar pattern (Fig. 1b). Particularly, the regional pairs with strong connectivity in the BOLD FC also presented strong connectivity in the SEEG FC. We next used Spearman's rank correlation to evaluate the similarity between the two FC matrices as the FC was not normally distributed. Before evaluating the correlation, we regressed out the Euclidean distance between pairs of regions from both matrices, as previous studies have reported associations between distance and FC[35,42]. We found that BOLD and SEEG white matter FC were significantly correlated ($r = 0.32$; $p_{FDR} < 0.001$) across all regional pairs (Fig. 1c). Finally, we observed that the Spearman's rank correlation between BOLD and SEEG white matter FC was significant in all frequency bands (median $r = 0.32$, $p_{FDR} < 0.001$; Fig. 1d).

### BOLD and SEEG white matter FC are correlated in every participant

Having demonstrated that BOLD and SEEG white matter FC were highly similar in a single participant, we next evaluated whether this phenomenon could be reproduced in the other 15 participants. By repeating the above procedure, we found that for each of the other 15 participants, BOLD and SEEG white matter FC were mostly significantly correlated across all regional pairs in all frequency bands after regressing out Euclidean distance from both FC matrices (Fig. 2). Overall, the median Spearman's rank correlation between BOLD and SEEG white matter FC across all participants was above $r = 0.19$ in all frequency bands (1–4 Hz: median $r = 0.19$; 4–8 Hz: median $r = 0.23$; 8–13 Hz: median $r = 0.21$; 13–30 Hz: median $r = 0.31$; 30–40 Hz: median $r = 0.31$; 40–70 Hz: median $r = 0.28$; 70–170 Hz: median $r = 0.25$. Figure 2 and Table S2). Evaluating the BOLD-SEEG correlation in white matter FC at both individual participant and individual frequency band levels, we observed that the correlations were significant with false discovery rate (FDR) corrected $p_{FDR} < 0.05$ in all seven frequency bands for 13 participants (See Table S2 for $r$ and $p_{FDR}$ of the correlations for all participants at each frequency band). In the remaining three participants, the correlations were significant with $p_{FDR} < 0.05$ in three, four, and six frequency bands, respectively. Notably, the FDR correction was used to account for the multiple comparisons across all the 16 participants and all frequency bands.

These results indicate that, as in gray matter[35], the BOLD FC also reflects the synchronization of intracranial electrophysiological signals (i.e., LFPs) in white matter, providing evidence for the electrophysiological basis of BOLD FC in white matter.

### Sensitivity analysis

We performed a series of additional analyses to validate the robustness of our results to methodological variation. Please refer to Supplementary Information for details of all results. Briefly, we demonstrated that our results were robust to the variation of parameters in fMRI processing, including analyzing data in native space (Fig. S4a and Table S4) rather than the standard space, using a bandpass filtering range of 0.01–0.08 Hz (Fig. S4b and Table S5) rather than 0.01–0.2 Hz, regressing out the global and CSF signals during preprocessing

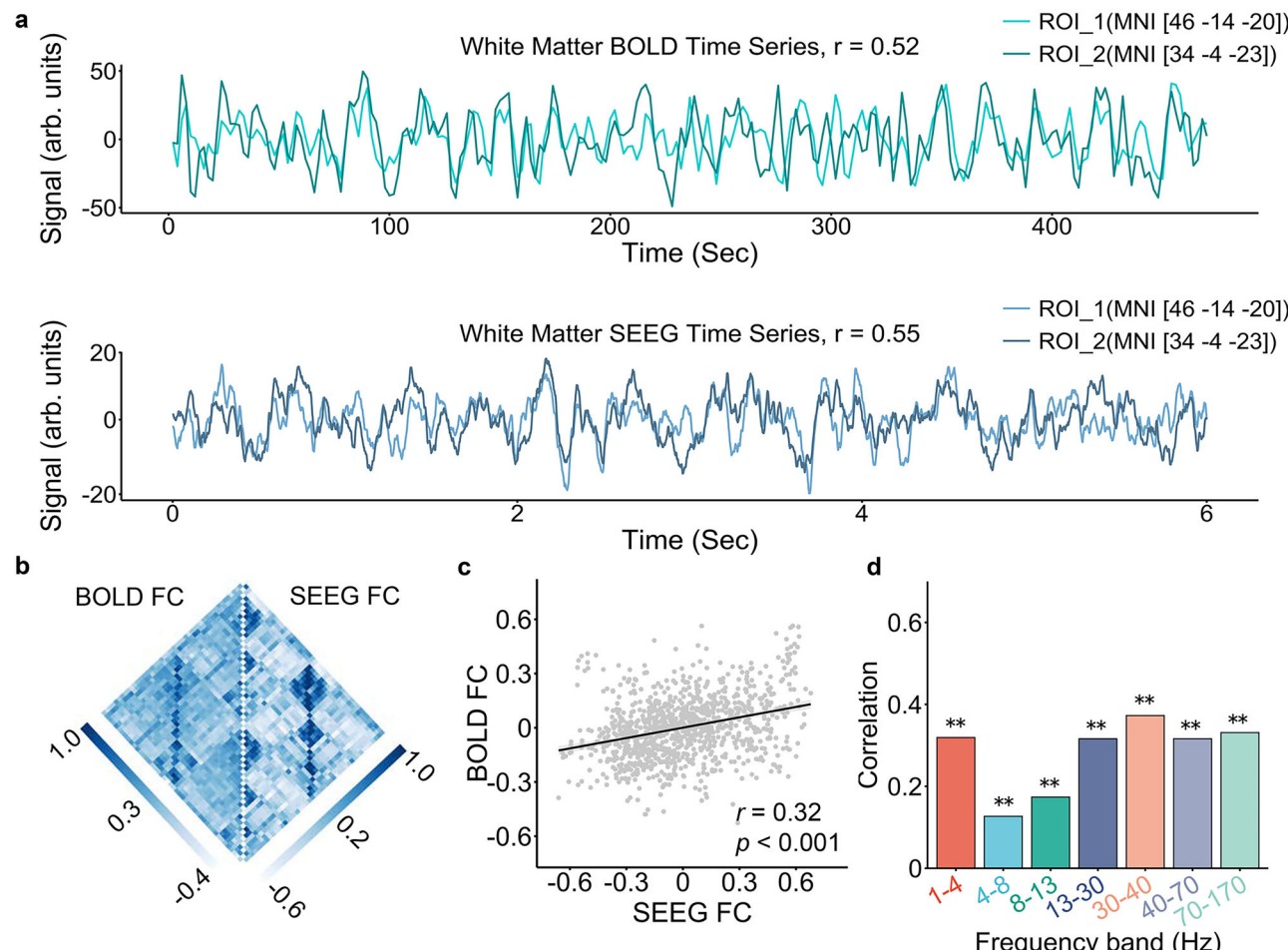

**Fig. 1 | BOLD and SEEG white matter FC are correlated in a single participant.**
**a** The time series of the two white matter contacts were highly synchronized in both BOLD and SEEG (1–4 Hz) data, suggesting an FC between the two contacts. The MNI coordinates of the two contacts were (46, −14, −20) and (34, −4, −23), respectively. We defined ROIs as spheres with the contact as the center and a radius of one voxel for the BOLD FC calculation. **b** Matrices of BOLD and SEEG (1–4 Hz) FC between every two white matter contacts. **c** BOLD and SEEG white matter FC were significantly correlated across all regional pairs in **b** (Spearman's rank correlation $r(1081) = 0.32$, $p_{FDR} = 1.2e-25$, two-sided). Each point indicated one pair of regions. The Euclidean distances between pairs of regions were regressed out from both BOLD and SEEG FC, and the acquired residuals were used to evaluate the correlation. The shaded envelope denotes the 95% confidence interval. **d** The correlation

between BOLD and SEEG FC was significant after regressing out the distances from both FC in all frequency bands (1–4 Hz: Spearman's rank correlation $r(1081) = 0.32$, $p_{FDR} = 1.2e-25$; 4–8 Hz: $r(1081) = 0.13$, $p_{FDR} = 4.1e-05$; 8-13 Hz: $r(1081) = 0.17$, $p_{FDR} = 1.7e-08$; 13-30 Hz: $r(1081) = 0.32$, $p_{FDR} = 2.7e-25$; 30–40 Hz: $r(1081) = 0.37$, $p_{FDR} = 6.5e-35$; 40–70 Hz: $r(1081) = 0.32$, $p_{FDR} = 2.7e-25$; 70–170 Hz: $r(1081) = 0.33$, $p_{FDR} = 1.2e-27$, all two-sided). The symbol (**) represents $p_{FDR} < 0.001$. Notably, while only one participant was analyzed here, the $p$ values were corrected with false discovery rate (FDR) to account for multiple comparisons across all participants and all frequency bands in this study. BOLD blood-oxygenation-level-dependent, SEEG stereotactic EEG, FC functional connectivity, ROI region of interest. Source data are provided as a Source Data file.

(Fig. S4c and Table S6), and using seven voxels neighbors (Fig. S5a and Table S7) or 27 voxels neighbors (Fig. S5b and Table S8) to define the ROIs for the BOLD FC calculation.

Our results were also robust to variation in parameters during SEEG data processing. We used 10 consecutive segments of the SEEG time series data with a length of 6 s, respectively, in the main analysis. Here, we tested 10 segments with a respective length of 4 s or 8 s, and found that the results were similar to our main results (see Fig. S6a and Table S9 for 4 s; see Fig. S6b and Table S10 for 8s). We used Pearson's correlation to evaluate the SEEG FC in the main analysis, and here we found that coherence-based SEEG FC also exhibited similar correlations with BOLD FC (Fig. S7 and Table S11).

**Structural connectivity constrains both BOLD and SEEG white matter FC**

Having demonstrated the underlying intracranial electrophysiological basis of white matter BOLD FC, we examined the structural connectivity basis of white matter FC. Using high-quality DSI data (-24 min

scanning), we reconstructed the whole-brain white matter fiber tracts for each participant. Next, we constructed a structural connectivity matrix (or structural network) by defining the network nodes as spheres centered at the coordinates of each contact and the network edges as the number of white matter tracts between two spheres.

Visual inspection indicated that the network matrix of white matter structural connectivity resembled that of BOLD white matter FC, typically presenting a higher FC with higher structural connectivity (sub1, Fig. 3a). Notably, the structural network was sparse, with many zero connections, suggesting that many FC emerged through indirect communication along structural connectivity. By quantitatively evaluating the Spearman's rank correlation across regional pairs with nonzero structural connections, we found the structural connectivity was significantly correlated with BOLD white matter FC after regressing out the distance from both matrices (sub1, $r = 0.16$, $p_{FDR} < 0.001$ Fig. 3b). Finally, our results showed that the structural connectivity correlated with BOLD white matter FC in all participants after regressing out the distance (median $r = 0.30$, Fig. 3c). Using FDR correction

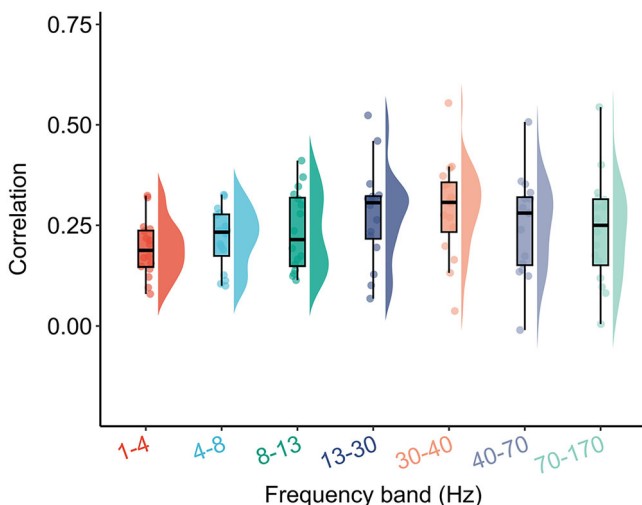

**Fig. 2 | The correlations between BOLD and SEEG white matter FC in all frequency bands for all 16 participants.** The Euclidean distances between pairs of regions were regressed out from both BOLD and SEEG FC before evaluating Spearman's rank correlations. There are 16 dots in each frequency band, representing the participants. The median correlations between BOLD and SEEG white matter FC across all participants were higher than $r = 0.19$ in all frequency bands (1–4 Hz: median $r = 0.19$; 4–8 Hz: median $r = 0.23$; 8–13 Hz: median $r = 0.21$; 13–30 Hz: median $r = 0.31$; 30–40 Hz: median $r = 0.31$; 40–70 Hz: median $r = 0.28$; 70–170 Hz: median $r = 0.25$). The correlations were significant with $p_{FDR} < 0.05$ (two-sided) in all seven frequency bands for 13 participants and the remaining three participants exhibited significant ($p_{FDR} < 0.05$, two-sided) correlations in three, four, and six frequency bands, respectively. See Table S2 for $r$ and $p_{FDR}$ of the correlations for all participants at each frequency band. Boxes denote the 25th to 75th percentile and the median line. Whiskers extend 1.5 times the interquartile range from the edges of the box. False discovery rate (FDR) correction was applied to account for multiple comparisons across all participants and all frequency bands. See Table S1 for the number of each participant's functional connections, which defined the sample size of the correlation analysis for each participant. BOLD blood-oxygenation-level-dependent, SEEG stereotactic EEG, FC functional connectivity. Source data are provided as a Source Data file.

across all participants, we found 14 participants showed significant ($p_{FDR} < 0.05$) correlations between structural connectivity and BOLD white matter FC, while the other 2 participants showed no significant correlation (See Table S12 for $r$ and $p_{FDR}$ for all participants).

We next evaluated the coupling between structural connectivity and SEEG white matter FC after regressing out the distance from both matrices. Similar to BOLD FC, the SEEG white matter FC (1–4 Hz) was also significantly correlated ($r = 0.36$, $p_{FDR} < 0.001$) with structural connectivity across regional pairs with nonzero structural connections (sub1, Fig. 3d, e). We also found that the median Spearman's rank correlation between SEEG white matter FC and structural connectivity across all participants was above $r = 0.22$ in each frequency band (1–4 Hz: median $r = 0.22$; 4–8 Hz: median $r = 0.23$; 8–13 Hz: median $r = 0.31$; 13–30 Hz: median $r = 0.34$; 30–40 Hz: median $r = 0.33$; 40–70 Hz: median $r = 0.33$; 70–170 Hz: median $r = 0.31$; Fig. 3f). Using FDR correction across all participants and all frequency bands, we observed that the correlations were significant with $p_{FDR} < 0.05$ in all seven frequency bands for eight participants (See Table S13 for $r$ and $p_{FDR}$ of the correlations for all participants at each frequency band). For the other seven participants, the correlations were significant ($p_{FDR} < 0.05$) in six frequency bands for four participants and were significant in five frequency bands for three participants. The remaining one participant (sub08) showed no significant correlation. Overall, these results suggest that the coupling between structural connectivity and SEEG FC is mostly replicable across different frequency bands and across individuals.

## Discussion

This study demonstrated the electrophysiological and structural basis of white matter BOLD FC using a multimodal dataset, including intracranial SEEG, resting-state fMRI, and DSI, from a group of 16 patients with drug-resistant epilepsy. We found that BOLD FC was correlated with SEEG FC in the white matter, and this result was consistent for each participant across a wide range of frequency bands. Moreover, white matter FC from both SEEG and BOLD fMRI was positively correlated with structural connectivity across all regional pairs, suggesting that anatomical structural connectivity constrains functional dynamics in white matter.

Our study builds on recent work showing that BOLD fMRI signal can be reliably detected in white matter tracts, which exhibit distinct responses to task loadings[4–10]. As in gray matter, the spontaneous fluctuations of BOLD signals synchronize between two spatially segregated regions in white matter, and this functional connectivity is organized into intrinsic functional networks[7,20–27]. Prior studies have demonstrated that BOLD FC in the white matter could be a potential neuromarker for both psychiatric and neurological disorders[28,31,32]. Our results provide the intracranial electrophysiological basis for these studies by showing that white matter BOLD FC reflects the synchronization of the underlying intracranial neural activity (i.e., LFPs) in the white matter.

Our results demonstrated that white matter BOLD FC was related to white matter electrophysiological FC, which was calculated using intracranial SEEG recordings. This association was significant across a wide range of frequency bands and for every participant, which robustly suggested the electrophysiological basis of BOLD FC in the white matter. This result is consistent with prior findings in gray matter[35,43]. In a landmark study, Logothetis et al. showed that the BOLD fMRI signal in the gray matter reflected the underlying LFPs[43], which laid the foundation of BOLD fMRI-based neuroscience studies. Recently, Betzel et al. demonstrated that the FC between cortical regions was similar between BOLD fMRI and intracranial ECoG data[35]. However, these findings were restricted to gray matter, whereas our work provides evidence that this is also true in white matter.

White matter has been ignored in functional brain studies for decades; however, our present work and a series of recent studies have consistently shown that white matter carries tract-specific, synchronized functional signals[7]. For example, recent studies have characterized the BOLD hemodynamic response function in the white matter, which displayed both task- and tract-specific patterns, distinct from that in the gray matter[5,10,15,16]. It has been also shown that the functional neuroplasticity in white matter tracts caused by motor learning could be detected using BOLD fMRI[17–19]. A recent study found that FC within white matter is higher than that in gray matter, although the white matter signal is weak[37]. Consistently, prior studies have shown that neuronal cell bodies exist in deeper white matter tissues[44], and that neurotransmitter vesicles are released directly into white matter[45], which could serve as the underlying neurobiological mechanism of white matter functional signals.

We found that white matter FC from both BOLD fMRI and SEEG data were highly correlated with structural connectivity, which was constructed using deterministic fiber tracking of white matter tracts with a high-quality DSI dataset. Prior studies have consistently found that the FC between gray matter regions is constrained by structural connectivity[38,46,47], and our results suggest that this is also true for FC between white matter regions. More importantly, regardless of whether using BOLD fMRI or intracranial SEEG, the white matter FC was consistently correlated with structural connectivity, underscoring the robustness of the observation. Our result is also consistent with recent work showing that distinct white matter bundles showed different BOLD activation patterns in both the resting state and in response to stimuli[11,12,20].

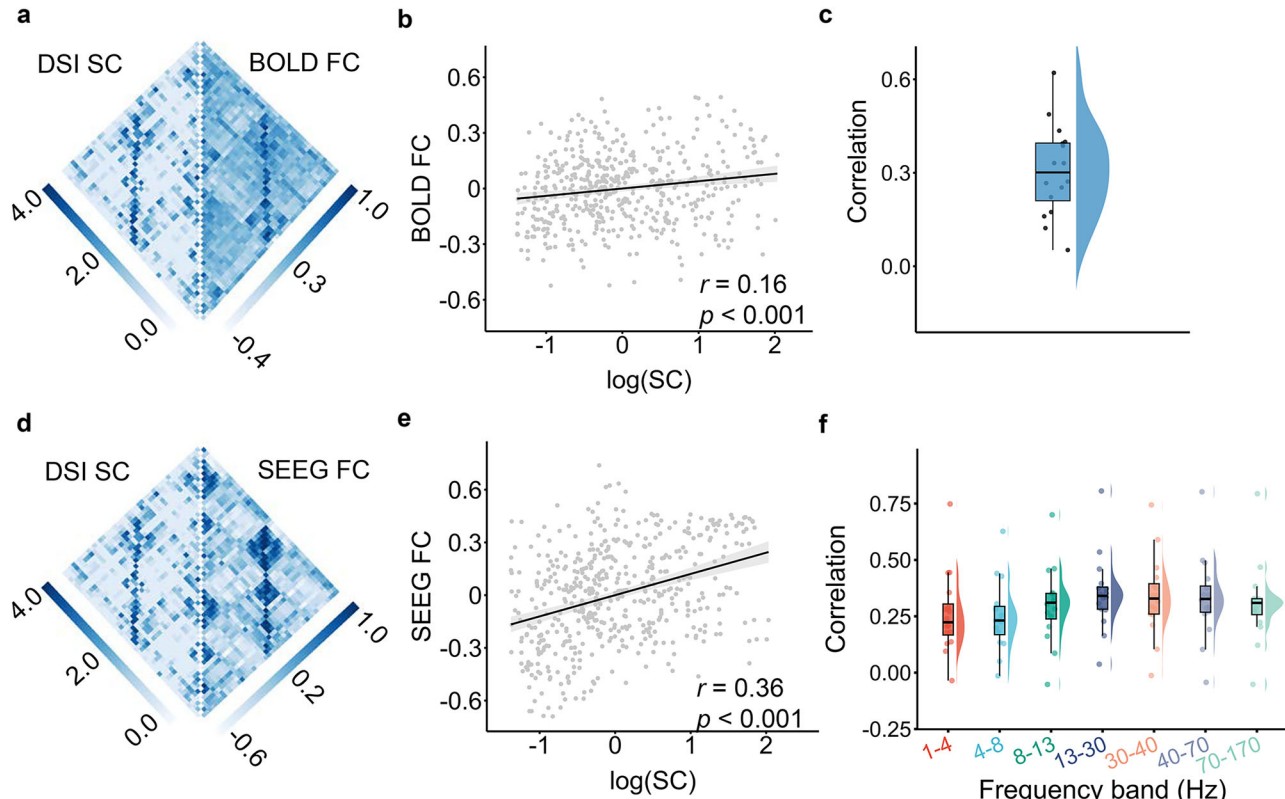

**Fig. 3 | White matter structural connectivity constrains both the BOLD and SEEG white matter FC. a** Matrices of structural connectivity and white matter BOLD FC in sub1. **b** Scatter plot of the correlation between structural connectivity and white matter BOLD FC using data from **a** (Spearman's rank correlation $r(511) = 0.16$, $p_{FDR} = 4.8e-04$, two-sided). The shaded envelope denotes the 95% confidence interval. **c** The correlations between structural connectivity and white matter BOLD FC for all 16 participants (median Spearman's rank correlation $r = 0.30$). See Table S12 for $r$ and $p_{FDR}$ for all participants. False discovery rate (FDR) correction was applied to account for multiple comparisons across all participants. **d** Matrices of structural connectivity and SEEG white matter FC with data filtered at 1–4 Hz in sub1. **e** Scatter plot of the correlation between structural connectivity and SEEG white matter FC using data from **d** (Spearman's rank correlation $r(511) = 0.36$, $p_{FDR} = 5.6e-15$, two-sided). The shaded envelope denotes the 95% confidence interval. **f** The median Spearman's rank correlations between structural connectivity and SEEG white matter FC across all 16 participants were higher than

$r = 0.22$ in each frequency band (1–4 Hz: median $r = 0.22$; 4–8 Hz: median $r = 0.23$; 8–13 Hz: median $r = 0.31$; 13–30 Hz: median $r = 0.34$; 30–40 Hz: median $r = 0.33$; 40–70 Hz: median $r = 0.33$; 70–170 Hz: median $r = 0.31$). See Table S13 for $r$ and $p_{FDR}$ for all participants and all frequency bands. Notably, the Euclidean distances between pairs of regions were regressed out from structural connectivity, SEEG FC, and BOLD FC before evaluating the correlations between matrices. FDR correction was applied to account for multiple comparisons across all participants and all frequency bands. In panels c and f, boxes denote the 25th to 75th percentile and the median line, and whiskers extend 1.5 times the interquartile range from the edges of the box. See Table S1 for the number of each participant's nonzero structural connections, which defined the sample size of the correlation analysis for each participant. BOLD blood-oxygenation-level-dependent, SEEG stereotactic EEG, FC functional connectivity, DSI diffusion spectrum imaging, SC structural connectivity. Source data are provided as a Source Data file.

This study has several potential limitations. First, we used a small sample of 16 patients with drug-resistant epilepsy. However, it should be noted that each participant underwent SEEG, resting-state BOLD fMRI, and DSI acquisitions. More importantly, our results could be replicated in almost each of the 16 participants. Therefore, we expect this result to be reproducible in a wide range of samples. Second, the SEEG recordings, which were clinically determined, were sparsely distributed in the white matter with a limited number of contacts, therefore only covering a small set of discrete brain regions. However, as each participant had contacts in different white matter areas, our data covered a large portion of the brain when we assembled the results of all the participants. Future studies should aggregate whole-brain data using SEEG recordings from a large sample of patients with drug-resistant epilepsy. Third, our BOLD fMRI and SEEG data were acquired in different sessions, which might have introduced additional variability. Future studies should evaluate the electrophysiological basis of white matter BOLD signals with the simultaneous acquisition of BOLD fMRI and SEEG. Fourth, the findings reported here were evaluated in medical-resistant epilepsy patients who had seizure severity requiring surgical intervention. Whether our results could be

generalized to healthy populations remains unclear. Finally, SEEG measures the LFPs of neural populations, and future studies may further evaluate whether the BOLD FC also reflects neuronal spiking using other techniques such as Utah arrays.

Notwithstanding these limitations, we provide evidence for the intracranial electrophysiological and structural basis of BOLD FC in white matter, which clearly suggests that white matter BOLD FC reflects the synchronization of neural activity and is constrained by underlying structural connectivity. Our data open an avenue for the origins and interpretations of BOLD signal synchronization in white matter and provide a foundation for the exploration of white matter BOLD FC as a potential neuromarker for both psychiatric and neurological disorders.

## Methods
### Participants
We included patients at the Xuanwu Hospital Capital Medical University with drug-resistant epilepsy who required SEEG monitoring to identify the precise origin of seizures. From a database of 84 participants, we selected 16 who had complete data with clinical SEEG

recordings as well as preoperative structural, diffusion, functional MRIs, and post-surgery X-ray CT. The participants were aged from 19 to 37 years, with a mean age of 28.2 years and a standard deviation (SD) of 4.9 years; this sample included nine males and seven females. See Table S1 for participants' information. All participants provided informed consent, and all study procedures were approved by the Institutional Review Boards of Xuanwu Hospital Capital Medical University.

## SEEG data acquisition

All participants underwent SEEG implantation using oblique approaches. The electrodes (ALCIS, Besancon, France) were placed using a ROSA robot system (ROSA, Medtech, Montpellier, France) based on preoperative enhanced MRI images to avoid vascular injury. The contacts of the SEEG electrodes were cylinders of platinum-iridium alloy, 2 mm in length and 0.8 mm in diameter. The center-to-center space between the contacts was 3.5 mm, and each electrode comprised 5–15 contacts. The contact locations were assessed using postoperative CT scans registered to the preoperative T1 images. The LFP was chronically recorded using a 256-channel Nicolet recording system (Natus Medical Incorporated, San Carlos, CA, United States). The sampling rate of the LFP recording was 2000 or 2048 Hz. Two experienced epileptologists interpreted the data and ensured there were no seizure events in the analyzed data.

## MRI acquisition and preprocessing

All MRI data, including structural MRI, BOLD fMRI, and DSI, were acquired using a GE Premier 3-T MRI scanner (General Electric Healthcare, Waukesha, WI, USA) with a 64-channel head coil at Xuanwu Hospital Capital Medical University.

**Structural MRI.** A magnetization-prepared, rapid acquisition gradient-echo (MPRAGE) T1-weighted image was acquired with the following parameters: TR, 2477 ms; TE, 2.69 ms; FOV, $256 \times 256$ mm$^2$; matrix, $256 \times 256$; 166 sagittal slices; slice thickness, 1 mm with no gap; and scanning duration, 6.8 min.

**BOLD fMRI.** Data were acquired with a gradient-echo echo planar imaging (EPI) sequence and the following parameters: TR, 2000 ms; TE, 30 ms; FOV, $224 \times 224$ mm$^2$; matrix, $64 \times 64$; slice thickness, 3.5 mm; voxel size, $3.5 \times 3.5 \times 3.5$ mm$^3$; the number of time points, 240; and scanning duration, 8 min.

**DSI.** Data acquisition was performed with 257 diffusion-weighted directions. The b-values ranged from 0 to 7000 s/mm$^2$. The hyperband acceleration factor was 2, and the other parameters were as follows: TR, 5548 ms; TE, 84.1 ms; voxel size, $2 \times 2 \times 2$ mm$^3$. The total acquisition time of the DSI sequence was ~24 min. A head stabilizer was inserted into the coil to prevent head motion.

## Electrode localization and SEEG data preprocessing

We obtained SEEG recordings from a total of 140 electrode shafts and 1474 contacts across all 16 participants. The Brainstorm (https://neuroimage.usc.edu/brainstorm/Tutorials/ECoG) pipeline was used for anatomical localization of the electrode contacts. We first used FreeSurfer[48] to reconstruct the brain surface from the T1-weighted images acquired prior to SEEG implantation. The CT images acquired after SEEG implantation were registered to the T1-weighted images using SPM12 (https://www.fil.ion.ucl.ac.uk/spm/), and the registration quality was checked using Brainstorm MRI Viewer[49]. Then, we manually labeled the location of each contact on the registered CT images based on the implantation scheme provided by neurosurgeons. Next, we normalized the T1 images to MNI space and applied the acquired warping transformation to register the native coordinates of the contacts to MNI space. According to an anatomical atlas (i.e., ASEG atlas[50]),

we identified which contacts were localized in the gray matter and which in the white matter. Among the 16 participants, 604 contacts were localized to the white matter. The following analyses were performed on white matter contacts.

Then, we preprocessed the recorded signals from white matter contacts using tools from FieldTrip (https://www.fieldtriptoolbox.org/) and a custom MATLAB pipeline. By visual inspection, we excluded the channels with (1) excessive noise from the power source, (2) no signal with a flat line, and (3) excessive flotation in the signal. We used the automatic artifact rejection pipeline from the FieldTrip toolbox to reject (i) jump, (ii) muscle, (iii) eye blink, and (iv) seizure spike artifacts (https://www.fieldtriptoolbox.org/tutorial/automatic_artifact_rejection/). We filtered the SEEG signals using a 0.5–300 Hz bandpass (Butterworth, third-order) and performed band-stop filtering to attenuate power-line noise (third-order Butterworth filter with band-stop between 49–61, 99–101, 149–151, 199–201, 249–251, 299–300 Hz). Finally, we re-referenced the signal from each channel/contact to the average signal across all the white matter channels.

## MRI data preprocessing

Structural and functional images were preprocessed using the fMRI-Prep toolbox[51] and eXtensible Connectivity Pipeline (XCP) Engine[52], which uses tools from FSL[53,54], AFNI[55], ANTs[56] and FreeSurfer (https://surfer.nmr.mgh.harvard.edu/). This pipeline includes (1) intensity non-uniformity correction and skull-stripping for T1-weighted images; (2) T1 segmentation into gray matter, white matter, and cerebrospinal fluid; (3) slice timing correction; (4) correction for susceptibility distortions induced by magnetic field inhomogeneity; (5) realignment of all volumes to a selected reference volume; (6) co-registration of the functional data to the structural image; (7) normalization to the MNI standard space; (8) de-meaning and removal of any linear trends; (9) regression of the 24 motion parameters, including six framewise estimates of motion, the derivatives of each of these six parameters, and quadratic terms of each of the six parameters and their derivatives; and (10) bandpass filtering with a passband between 0.01–0.2 Hz. We removed the initial five volumes from the data. Finally, the data were resampled to 2 mm isotropic resolution to facilitate the analysis.

The DSI dataset was preprocessed using the QSIPrep pipeline[57], which is a pipeline toolbox for diffusion MRI data processing based on other toolboxes, such as FSL[53,54] and ANTs[56]. The preprocessing steps included: (1) transforming all images and bvecs into a consistent orientation system; (2) denoising the images using Marchenko-Pastur (MP)-PCA, Gibbs unringing, and B1 bias correction; (3) normalizing the intensity across all $b = 0$ images; (4) estimating and correcting head motion using the SHORELine technique, which first aligned non-b0 images to b0 images and then used a leave-one-out procedure to create target signal images and register the left-out image and the corresponding vector to the target; (5) generating a $b = 0$ template image and registering all diffusion-weighted images to the template; and (6) registering all images to the individual T1-weighted images.

## Calculation of BOLD and SEEG white matter FC

The average time series of white matter BOLD signals were extracted from a 3 mm radius sphere, which comprised 19 neighboring voxels adjacent to the surface and edge, at each contact location within the white matter. White matter BOLD FC was computed as the Pearson's correlation between the time series from each pair of white matter contacts. For SEEG data, we applied bandpass filtering to filter the signals into seven different frequency bands (1–4 Hz, 4–8 Hz, 8–13 Hz, 13–30 Hz, 30–40 Hz, 40–70 Hz, and 70–170 Hz). White matter SEEG FC was estimated as Pearson's correlation between the SEEG signals from each pair of white matter contacts. Finally, we acquired white matter BOLD FC and SEEG FC matrices for each participant. The BOLD FC and SEEG FC matrices shared the same brain regions, which were defined

by the positions of the contacts, making the two matrices comparable. For each participant, we selected one SEEG time series with a length of 60 s and divided it into ten segments, each containing a window size of 6 s. We calculated the average SEEG time series of the ten segments to increase the stability and signal-to-noise ratio of the SEEG data. Finally, we computed the white matter SEEG FC based on this average time series.

### Structural connectome construction with DSI
We constructed a structural connectivity matrix for each participant using DSI Studio (http://dsi-studio.labsolver.org). We first compared the orientation in the b-table to the population-averaged template[58] and quantified the restricted diffusion[59]. A generalized q-sampling imaging approach was adopted to generate an orientation distribution function (ODF) map with a diffusion sampling length ratio of 1.25. Based on the ODF, we reconstructed the whole-brain white matter tracts using a quantitative anisotropy-based deterministic fiber tracking algorithm[60]. We used a randomized quantitative anisotropy threshold with the following tracking parameters: angular threshold, 90 °; step length, 0.5 mm; tracking length, 2–350 mm; the total number of tracts, 2,000,000. To construct the white matter connectivity matrix, we defined ROIs as spheres with a radius of one voxel, which included 7 neighboring voxels adjacent to surface, surrounding the contacts in the white matter. The strength of each connectivity was defined as the number of interconnecting fiber tracts between each ROI pair.

### Reporting summary
Further information on research design is available in the Nature Portfolio Reporting Summary linked to this article.

## Data availability
All the data required to reproduce our findings have been made publicly available (https://github.com/CuiLabCIBR/IEEGwmFC/tree/main/data), including BOLD and SEEG functional connectivity, structural connectivity, and the distance matrix for all the participants. The relevant data for visualizing the figures are provided as Source Data files. Raw data is available from the corresponding authors upon request. Source data are provided with this paper.

## Code availability
All analysis codes are available here: https://github.com/CuiLabCIBR/IEEGwmFC, with a detailed explanation at the following link: https://github.com/CuiLabCIBR/IEEGwmFC/wiki.

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

## Acknowledgements

This work is supported by the STI 2030-Major Projects (2022ZD0211300 to Z.C.), the National Natural Science Foundation of China (82030037 and 81871009 to G.Z.), the Chinese Red Cross Foundation 'Six Walnuts·National Brain Nutrition Research Fund' (to Y.H.), Beijing Nova Program (Z211100002121002 to Z.C.), CIBR funds (to Z.C.), the Translational and Application Project of Brain-inspired and Network Neuroscience on Brain Disorders (to G.Z.), and Beijing Municipal Health Commission (11000022T000000444685 to G.Z.). We thank Dr. Meichen Yu for the helpful discussion.

## Author contributions

Z.C., G.Z., Y.H., and P.-H.W. designed the study. Z.C. and G.Z. supervised the project. G.Z., P.-H.W., D.C., and Y.F.Y. acquired the multimodal data for epilepsy patients. L.X. and Y.H. performed the data preprocessing with support from P.-H.W., D.C., and Y.F.Y. Y.H. and L.X. performed the analyses and visualizations, with supervision from Z.C. and support from Q.F., G.W., X.J., Y.Y.Y., and W.S. Y.H., L.X., P.-H.W., and Z.C. wrote the manuscript. All authors reviewed and edited the manuscript.

## Competing interests

The authors declare no competing interests.
