## [Peer Review File · Nature Communications]

Intracranial electrophysiological and structural basis of BOLD functional connectivity in human brain white matterReviewer #1 (Remarks to the Author):

This manuscript reports a multimodal study of the human brain that explores electrophysiological and structural basis of BOLD functional connectivity in white matter. As white matter BOLD signals are being increasingly recognized to carry neural activity information and play important roles in a wide variety of neurological and psychiatric disorders, studies of this kind could have great impact to the research field. Overall the study is highly novel and the findings are impressive, which potentially provides a very first direct evidence that white matter BOLD signals are underlied by neurophysiological processes. That being said, the reviewer has identified a number of weaknesses that the authors need to address.

1. Abstract: In "suggesting anatomical fiber tracts supports the functional synchronization in white matter", "supports" should be "support" or more preferably "underlie".

2. Introduction: The problem is well posed with the motivation of this work clearly articulated. There are some minor language problems that need to improve. For example:

a). The statement "which were organized into three layers with distinct levels of correlation with gray matter FC" is confusing. It meant "organized into three layers with distinct levels of correlation with cortical gray matter"?

b). In "these networks were organized into two groupings with anti-correlated connectivity", "groupings" should be "groups".

c). What the sentence "The white matter BOLD FC aligns with a specific structure of anatomical white matter tracts" refers to specifically is not quite clear.

d). Add "or merely a vascular phenomenon" to the end of the sentence "it is still unclear whether the white matter BOLD FC reflects the underlying neural synchronization of intracranial electrophysiological signals in white matter".

e). In "providing an opportunity to recode the LFPs in white matter tissues", should "recode" be "record"?

f). In "Recently, Andrew et al. revealed that white matter FC is stronger than gray matter FC although the white matter signals were weaker using SEEG data", "Andrew" should be "Revell". Also the statement "However, this study did not examine the electrophysiological basis of BOLD FC in white matter" is misleading. Did you mean "However, this study did not seek correlations between electrophysiological signals with BOLD FC in white matter"?

3. Results:

a). The results from this work crucially depend on the experimental settings, particularly on the distribution of the depth electrodes and contacts. As presented in the current form, however, the relevant information is missing, which undermines reader's confidences in the reported findings. Detailed placement of the electrodes and contacts should be described, preferably with an illustrative figure showing a typical case. The average length of the electrodes as reported was 6 mm, which is much shorter than that reported in other studies (~10 contacts with >2 mm spacing). See "Stereoencephalography: Indication and Efficacy" (Koji IIDA et al. in PMC5566696), in which depth electrodes used for SEEG typically have 4-18 contacts spaced 2-10 mm apart with a diameter of 1 mm or less.

b). The short electrode span in this study brought a concern that whether the observed white matter BOLD FC was principally driven by spatially adjacent contacts. This should be carefully examined both qualitatively and quantitatively to justify the reported findings. One option is to order the distances between all the pairs of contacts included in the correlation analysis, and compute the amount of variances in BOLD FC explainable by SEEG FC by progressively excluding the pairs of contacts with short distances.

c). Change "In the final one participant" to "In the remaining participant".

d). Provide a reference for "These results indicated, as in gray matter, the BOLD FC also reflected the synchronization of intracranial electrophysiological signals".

e). The statement "suggesting the BOLD FC in white matter may reflect more electrophysiological signals in middle-frequency bands" does not necessarily hold.

f). Change "To our understanding" to "To our knowledge",

g). "Regarding BOLD FC, the SEEG white matter FC (1-4 Hz) was significantly correlated ($r = 0.46$, $p < 0.001$) with structural connectivity..." Did it mean "Similar to BOLD FC"?

4. Discussion: This work acquired BOLD and sEEG signals from different sessions, which could contribute additional variability to the regression analysis due to the nonstationary nature of brain activity. Also, the findings reported here are limited to epilepsy patients, which may not be generalizable to healthy subjects. Both points should be included as limitations.

There are many language/expression issues in this section as well. The following is only a brief list of obvious ones:

- a). Unify "electrophysiology FC" and "electrophysiological FC"
- b). Change "electrophysiology basis" to "electrophysiological basis".
- c). Remove the comma (,) from "Logothetis et al., showed"
- d). The statement "our present work and a series of recent studies have consistently shown that white matter carries both functional signals and the synchronization of these signals" can be better rephrased as "our present work and a series of recent studies have consistently shown that white matter carries tract-specific, synchronized functional signals".
- e). Change "neurobiology mechanism" to "neurobiological mechanism".
- f). Change "we expected this result should be reproduced in a wide range of samples" to "we expect this result to be reproducible in a wide range of samples".
- g). Change "covering a small part of the entire brain" to "covering a small set of discrete brain regions".

To sum it up, the work presented is highly novel and of significant interest to the research community. There are needs for relevant descriptive details of the experimental settings and more in-depth analysis of the findings reported, in addition to thorough language improvements.

Reviewer #2 (Remarks to the Author):

Huang and colleagues demonstrate that white matter functional connectivity, estimated by sEEG at multiple frequency bands, is correlated with the BOLD functional connectivity estimated through resting-state fMRI. They additionally demonstrate that structural connectivity estimated by diffusion tractography is significantly correlated with both BOLD and sEEG functional connectivity. The authors argue that these correlations imply an electrophysiological basis for white matter BOLD signal, and that this electrophysiology is constrained by the structural connectivity.

Methods

Subjects

They included 16 patients with drug resistant epilepsy who had stereo EEG implantation - It is not clear where in the white matter exactly these electrodes were located, should include a general heatmap that shows the sampling of these regions.

Electrode Co-Registration

The authors used an in-house co-registration pipeline. It is not clear, however, whether the final analyses, both in the sEEG and the fMRI were done in native space or MNI space. This has to be specified given it is known that registration to MNI space during the coregistration process can cause stereotactic electrodes to bend, leading to imprecise localization of the electrodes.

sEEG Processing

The authors broke down the sEEG signal into small time segments and then computed the Pearson correlation between the smaller segments to improve the SNR. This is a valid approach. They also ensured that no ictal activity was present in the recordings, which eliminates ictal activity as a confound from the findings. They computed their sEEG functional connectivity across a wide range of frequency bands ranging from 1-170Hz.

fMRI Preprocessing and Functional Connectivity Estimation

The authors utilized state of the art pre-processing pipelines (namely fMRIPrep and XCPengine), which allows for a standard and reproducible pre-processing of the data. It is not clear to me why they bandpassed their fMRI signal between 0.01-0.2, since the common range utilized, and the

one demonstrated to allow for appropriate filtering by Satterthwaite et. al. 2014, is 0.01-0.08. This needs to be justified further.

Structural Imaging Processing

The structural imaging processing was appropriate.

Main Results - Major Concerns

The findings demonstrate that the white matter BOLD and sEEG signals are significantly correlated across the selected frequency bands. The authors report that this is the case for many of the subjects at both an individual subject level and across frequency bands. However, the authors do not report the specific p-values for each subject, and it does not appear that they performed any correction for multiple comparisons across the manuscript. In the simple case of a Bonferroni procedure, there were 16 patients and 7 frequency bands, which would require a significant p-value of 0.0004 to imply significance, but the authors report at most $p < 0.001$ (more conservative approaches such as a FWER correction could also be applied). Multiple comparison correction is omitted throughout the entirety of the manuscript, and the lack of transparency on the p-values reported makes the results difficult to interpret.

The main finding of this study is a "statistically significant" correlation between white matter sEEG FC and BOLD FC. Additionally, there is also a correlation between a structural connectome of the same white matter regions and the sEEG and BOLD FC. The authors argue that this correlation implies an electrophysiological basis for the white matter BOLD signal. While this is potentially true, the results of the study, and the experimental design, do not answer this question. One of the studies that the authors reference is the one by Betzel et. al. 2019, where a correlation was found between cortical gray matter nodes in sEEG and BOLD. However, they seem to ignore the rest of the manuscript, where Betzel et. al. demonstrate that distance between regions has a very large

influence in this correlation. In fact, one of the supplementary figures in that manuscript (supplementary figure 4), demonstrates the significant impact of Euclidean distance between regions on the the correlation between the sEEG functional connectivity and other metrics. Therefore, these results are very likely to be mostly driven by a distance effect between the white matter regions, and not truly by electrophysiological signals or structural tractography, or at least that is my intuition without seeing an analysis being done to convince me otherwise. A good approach to test this would be to regress the Euclidean distance between the white matter regions and assess if the sEEG FC still predicts a significant amount of variance in the BOLD FC.

Because the purpose of this study is to demonstrate a "novel" correlation in the white matter between sEEG and BOLD, there has to be significantly more rigor in testing what is driving this correlation. For example, there should be appropriate null models to compare the correlations (e.g. is the correlation between WM sEEG FC and BOLD FC higher than that of WM sEEG FC and the Euclidean distance matrix between the electrodes?). There should also be additional sensitivity analyses: how does fMRI preprocessing affect these correlations, does the window size matter for the sEEG connectivity, does the radius of the electrode ROI affect the findings, do other correlation metrics more commonly used in sEEG FC such as coherence provide different findings?

Overall the manuscript addresses an important problem, but lacks the scientific rigor necessary to ensure that the results support the conclusions the authors make on them, and to ensure that the findings are robust and reproducible.

General Summary of Impressions

Huang and colleagues demonstrate that white matter functional connectivity, estimated by sEEG at multiple frequency bands, is correlated with the BOLD functional connectivity estimated through resting-state fMRI. They additionally demonstrate that structural connectivity estimated by diffusion tractography is significantly correlated with BOLD and sEEG functional connectivity. While the findings are novel, there are multiple serious theoretical and methodological considerations that decrease the quality and interpretation of the findings. First, this is an exploratory study, therefore care has to be taken in correcting for

spurious effects and multiple comparisons. The authors did not do this throughout the manuscript. Second, the authors are trying to establish the existence of a correlation between WM sEEG FC and WM BOLD FC, therefore significantly more rigor (appropriate null models, sensitivity analyses, etc.) is required to ensure validity of the results. Specifically, it is well known that these correlation metrics are highly influenced by the distance between the nodes (either voxels, ROIs, or electrodes), therefore these results are very likely to be mostly driven by a distance effect between the white matter regions, and not truly by electrophysiological signals or structural tractography. Overall the manuscript addresses an important problem, but lacks the scientific rigor necessary to ensure that the results support the conclusions the authors make on them, and to ensure that the findings are reproducible.

Reviewer #3 (Remarks to the Author):

The authors present an important study comparing intracranial electrophysiological and BOLD fMRI data in white matter. The results provide novel evidence to further support functional MRI in white matter, an area I know well. While the study overall is commendable, there are some outstanding issues and questions to address:

Major:

1) It is critically important to add a caveat in the discussion that these results are in patients who have seizure severity requiring surgical intervention - and therefore the impacts of this likely affect the underlying physiological relationship relative to individuals who do not experience seizures (ie, generalizability remains to be demonstrated).

2) Please explain why Spearman's correlations were used rather than Pearson's correlations. Is there a rationale for a non-parametric approach? It would be good to have also run Pearson's to know if this was insensitive and why?

Minor:

The cited literature is not up-to-date. A fair amount of recent studies should be cited to summarize recent advances in both characterizing the white matter HRF and sensitivity to white matter neuroplasticity to name a few areas that demonstrate the growing validity of white matter fMRI.

Reviewer #1 (Remarks to the Author):

This manuscript reports a multimodal study of the human brain that explores electrophysiological and structural basis of BOLD functional connectivity in white matter. As white matter BOLD signals are being increasingly recognized to carry neural activity information and play important roles in a wide variety of neurological and psychiatric disorders, studies of this kind could have great impact to the research field. Overall the study is highly novel and the findings are impressive, which potentially provides a very first direct evidence that white matter BOLD signals are underlied by neurophysiological processes. That being said, the reviewer has identified a number of weaknesses that the authors need to address.

We thank the reviewer for the positive appraisal of our work and for his or her insightful comments. As described below, we have incorporated the reviewer's valuable comments into the revised manuscript.

1. *Abstract: In "suggesting anatomical fiber tracts supports the functional synchronization in white matter", "supports" should be "support" or more preferably "underlie".*

Thank you for pointing this out. We have corrected this in the revised manuscript accordingly (Abstract).

..., suggesting that anatomical fiber tracts underlie the functional synchronization in white matter.

2. *Introduction: The problem is well posed with the motivation of this work clearly articulated. There are some minor language problems that need to improve. For example:*

a). *The statement "which were organized into three layers with distinct levels of correlation with gray matter FC" is confusing. It meant "organized into three layers with distinct levels of correlation with cortical gray matter"?*

b). *In "these networks were organized into two groupings with anti-correlated connectivity", "groupings" should be "groups".*

c). *What the sentence "The white matter BOLD FC aligns with a specific structure of anatomical white matter tracts" refers to specifically is not quite clear.*

d). *Add "or merely a vascular phenomenon" to the end of the sentence "it is still unclear whether the white matter BOLD FC reflects the underlying neural synchronization of intracranial electrophysiological signals in white matter".*

e). *In "providing an opportunity to recode the LFPs in white matter tissues", should "recode" be "record"?*

f). *In "Recently, Andrew et al. revealed that white matter FC is stronger than gray matter FC although the white matter signals were weaker using SEEG data", "Andrew" should be "Revell". Also the statement "However, this study did not examine the*

electrophysiological basis of BOLD FC in white matter" is misleading. Did you mean "However, this study did not seek correlations between electrophysiological signals with BOLD FC in white matter"?

We appreciate you pointing out these language problems. We have edited these text parts individually, as suggested, and the manuscript was edited by a native speaker.

a) Yes. We have revised this sentence according to your suggestion (Page 3, Para. 2):

For example, Peer et al. parcellated the white matter into 12 symmetrical functional networks, which were organized into three layers with distinct levels of correlation with cortical gray matter functional networks²⁰.

b) We have corrected it in the revised manuscript (Page 3, Para. 2):

..., and that these networks were organized into two groups with anti-correlated connectivity²¹.

c) We apologize for the confusion. We tried to express that the structure of white matter functional networks is constrained by anatomical white matter tracts. We have revised this sentence in the revised manuscript accordingly (Page 3, Para. 2):

The white matter BOLD FC is constrained by the structure of anatomical white matter tracts^{20, 21} and ...

d) Thank you for this suggestion. We have revised the manuscript accordingly (Page 4, Para. 1)

However, it remains still unclear whether the white matter BOLD FC reflects the underlying neural synchronization of intracranial electrophysiological signals in the white matter or merely a vascular phenomenon.

e) Thank you. We have corrected it (Page 4, Para. 2).

..., providing an opportunity to record LFPs in white matter tissues.

f) Thank you for this suggestion. We have revised the manuscript accordingly (Page 4, Para. 2).

Recently, Revell et al. revealed that white matter FC is stronger than gray matter FC, although white matter signals were weaker in SEEG data³⁴. However, this study did not seek correlations between electrophysiological signals with BOLD FC in white matter.

3. Results:

a). *The results from this work crucially depend on the experimental settings, particularly on the distribution of the depth electrodes and contacts. As presented in the current form, however, the relevant information is missing, which undermines reader's confidences in the reported findings. Detailed placement of the electrodes and contacts should be described, preferably with an illustrative figure showing a typical case. The average length of the electrodes as reported was 6 mm, which is much*

shorter than that reported in other studies (~10 contacts with >2 mm spacing). See "Stereoencephalography: Indication and Efficacy" (Koji IIDA et al. in PMC5566696), in which depth electrodes used for SEEG typically have 4-18 contacts spaced 2-10 mm apart with a diameter of 1 mm or less.

Thank you for this suggestion. We agree that the localization of the electrodes and contacts is important for understanding the results. In the revised manuscript, we have provided an illustrative figure showing the electrodes of one participant (sub1) and the aggregation of all participants' electrodes in the new **Fig. S1**. We found that the contacts were mainly localized in the temporal, frontal, and parieto-temporal areas. To localize the precise anatomical location of these contacts, we used the a priori White Matter Parcellation Map (WMPM) atlas (Mori et al., 2008; Oishi et al., 2008), which includes 68 regions with 50 core and 18 peripheral white matter areas. We have added the precise localization of the contacts in both the main results and Supplementary Materials.

Main Results (Page 5, Para. 2):

Using an a priori White Matter Parcellation Map (WMPM) atlas^{36,37}, we found that white matter contacts were mainly localized in the temporal, frontal, and parieto-temporal areas (**Fig. S1a** and **Fig. S1b**, See **Supplementary Materials** for details).

Supplementary Materials (Page 1, Para. 1):

To localize the precise anatomical locations of the white matter contacts, we used the a priori White Matter Parcellation Map (WMPM) atlas^{1,2}, which includes 68 regions with 50 core and 18 peripheral white matter areas. First, we depicted a single participant (sub1, **Fig. S1a**) in the volume space. For better visualization, we projected all electrodes onto the same slice. We found a total of 47 white matter contacts for this participant, which were mainly localized in the temporal blade (28 contacts), inferior frontal blade (7 contacts), and parieto-temporal blade (5 contacts) (**Fig. S1a**). Next, we aggregated all 604 white matter contacts from all participants (**Fig. S1b**). Using the WMPM atlas, we found that the contacts were mainly localized in the temporal blade (288 contacts), frontal blade (122 contacts), parieto-temporal blade (54 contacts), anterior corona radiata (21 contacts), sagittal stratum (20 contacts), posterior thalamic radiation (15 contacts).

Fig. S1 Distribution of the contacts of SEEG implanted electrodes. a) The electrodes for one participant (sub1) in volume space. To facilitate visualization, we projected all electrodes to one slice of the brain. **b)** The aggregation of all participants' electrodes visualized in surface space. Red color indicates contacts in gray matter and green color indicates contacts in white matter. There are 604 white matter contacts in total. SEEG: stereotactic EEG.

We apologize for the confusion regarding the electrode length and speculate this information comes from the Introduction part. We originally wanted to express that the typical average center-to-center distance between adjacent contacts is 6 mm in the field, rather than discussing about the between-contact distance in our study. As we reported in the method section ("SEEG data acquisition"), the SEEG electrodes (ALCIS, Besancon, France) in our study typically included 5-15 contacts and each contact was 2 mm in length and 0.8 mm in diameter. The center-to-center space between the contacts was 3.5 mm. We have rephrased the sentence in the Introduction according to the reference (Iida and Otsubo, 2017) you provided (Page 4, Para. 2).

In contrast to ECoG, SEEG electrodes typically penetrate the brain through white matter and have 4-18 contacts with a center-to-center space between two adjacent contacts ranging from 2–10 mm³³, providing an opportunity to record LFPs in white matter tissues.

b). The short electrode span in this study brought a concern that whether the observed white matter BOLD FC was principally driven by spatially adjacent contacts. This should be carefully examined both qualitatively and quantitatively to justify the reported findings. One option is to order the distances between all the pairs of contacts included in the correlation analysis, and compute the amount of variances in BOLD FC explainable by SEEG FC by progressively excluding the pairs of contacts with short distances.

This is an important point. We agree that the spatial distance between two contacts

could affect the estimation of white matter BOLD FC. We appreciate the recommended analysis and thought a simpler way is controlling for the distance in the correlation analysis between the two FC matrices, which has been used in prior literatures (Betzel et al., 2019; Hansen et al., 2022). We computed the Euclidean distance between each pair of contacts for using the following formula: distance = $\sqrt{(x_1 - y_1)^2 + (x_2 - y_2)^2 + (x_3 - y_3)^2}$, where (x_1, x_2, x_3) and (y_1, y_2, y_3) are the coordinates of the centers of the two respective contacts. We calculated a distance matrix for each participant with each element in the matrix quantifying the Euclidean distance between each pair of contacts. Next, we regressed out the distance matrix from both the BOLD FC and SEEG FC matrices using a linear model, and then evaluated the Spearman's rank correlation between the two residual matrices. We found the correlations between BOLD and SEEG white matter FC across all participants was above $r = 0.19$ in all frequency bands (1–4 Hz: median $r = 0.19$; 4–8 Hz: median $r = 0.23$; 8–13 Hz: median $r = 0.21$; 13–30 Hz: median $r = 0.31$; 30–40 Hz: median $r = 0.31$; 40–70 Hz: median $r = 0.28$; 70–170 Hz: median $r = 0.25$. **New Fig. 2**; See **Table S2** for accurate r and p_{FDR} values). Of all the 112 correlations for all participants and all frequency bands, 104 were significant ($p_{FDR} < 0.05$) after FDR correction (**Table S2**). Now, we used this result with distance regressed out as our main result in the revised manuscript (Page 8, Para. 1).

By repeating the above procedure, we found that for each of the other 15 participants, BOLD and SEEG white matter FC were mostly significantly correlated across all regional pairs in all frequency bands after regressing out Euclidean distance from both FC matrices (**Fig. 2**). Overall, the median Spearman's rank correlation between BOLD and SEEG white matter FC across all participants was above $r = 0.19$ in all frequency bands (1–4 Hz: median $r = 0.19$; 4–8 Hz: median $r = 0.23$; 8–13 Hz: median $r = 0.21$; 13–30 Hz: median $r = 0.31$; 30–40 Hz: median $r = 0.31$; 40–70 Hz: median $r = 0.28$; 70–170 Hz: median $r = 0.25$. **Fig. 2** and **Table S2**). Evaluating the BOLD-SEEG correlation in white matter FC at both individual participant and individual frequency band levels, we observed that the correlations were significant with false discovery rate (FDR) corrected $p_{FDR} < 0.05$ in all seven frequency bands for 13 participants (See **Table S2** for r and p_{FDR} of the correlations for all participants at each frequency band). In the remaining three participants, the correlations were significant with $p_{FDR} < 0.05$ in three, four, and six frequency bands, respectively.

New Fig. 2 The correlations between BOLD and SEEG white matter FC in all frequency bands for all the 16 participants. The Euclidean distances between pairs of regions were regressed out from both BOLD and SEEG FC before evaluating the correlations. There are 16 dots in each frequency band, representing the participants. The median correlations between BOLD and SEEG white matter FC across all participants were higher than $r = 0.19$ in all frequency bands (1-4 Hz: median $r = 0.19$; 4-8 Hz: median $r = 0.23$; 8-13 Hz: median $r = 0.21$; 13-30 Hz: median $r = 0.31$; 30-40 Hz: median $r = 0.31$; 40-70 Hz: median $r = 0.28$; 70-170 Hz: median $r = 0.25$). The correlations were significant with $p_{FDR} < 0.05$ in all seven frequency bands for 13 participants and the remaining three participants exhibited significant ($p_{FDR} < 0.05$) correlations in three, four, and six frequency bands, respectively. See **Table S2** for r and p_{FDR} of the correlations for all participants at each frequency band. BOLD: blood-oxygenation-level-dependent; SEEG: stereotactic EEG; FC: functional connectivity.

- c). Change "In the final one participant" to "In the remaining participant".
- d). Provide a reference for "These results indicated, as in gray matter, the BOLD FC also reflected the synchronization of intracranial electrophysiological signals".
- e). The statement "suggesting the BOLD FC in white matter may reflect more electrophysiological signals in middle-frequency bands" does not necessarily hold.
- f). Change "To our understanding" to "To our knowledge",
- g). "Regarding BOLD FC, the SEEG white matter FC (1-4 Hz) was significantly correlated ($r = 0.46$, $p < 0.001$) with structural connectivity..." Did it mean "Similar to BOLD FC"?

We appreciate your comments regarding language. We have addressed these points accordingly and have edited the language throughout the paper.

c) This was edited according to the suggestion (Page 8, Para. 1).

In the **remaining** three participants, the correlations were significant with $p_{FDR} < 0.05$ in three, four, and six frequency bands, respectively.

d) We have added the reference in the revised manuscript (Page 8, Para. 2).

These results indicate that, as in gray matter³², the BOLD FC also reflects the synchronization of intracranial electrophysiological signals (i.e., LFPs) in white matter.

Reference (32): R. F. Betzel et al., Structural, geometric and genetic factors predict interregional brain connectivity patterns probed by electrocorticography. *Nat Biomed Eng* 3,

902-916 (2019).

e) We agree and have deleted this sentence from the results section and associated content in the discussion section of the revised manuscript.

f) Thank you, we have changed it accordingly (Page 8, Para 2).

To our knowledge, this is the first report of the association between BOLD and SEEG FC in white matter, providing evidence for the electrophysiological basis of BOLD FC in white matter.

g) Thank you for pointing this out. We have corrected it (Page 11, Para. 2).

Similar to BOLD FC, the SEEG white matter FC (1–4 Hz) was also significantly correlated ($r = 0.36$, $p_{\text{FDR}} < 0.001$) with structural connectivity across regional pairs with non-zero structural connections (sub1, **Fig. 3d and 3e**).

4. Discussion: This work acquired BOLD and SEEG signals from different sessions, which could contribute additional variability to the regression analysis due to the nonstationary nature of brain activity. Also, the findings reported here are limited to epilepsy patients, which may not be generalizable to healthy subjects. Both points should be included as limitations.

Thank you for these comments. We have added both limitations to the revised manuscript (Page 15, Para. 2).

Third, our BOLD fMRI and SEEG data were acquired in different sessions, which might have introduced additional variability. Future studies should evaluate the electrophysiological basis of white matter BOLD signals with the simultaneous acquisition of BOLD fMRI and SEEG. Fourth, the findings reported here were evaluated in medical-resistant epilepsy patients who had seizure severity requiring surgical intervention. Whether our results could be generalized to healthy populations remains unclear.

There are many language/expression issues in this section as well. The following is only a brief list of obvious ones:

- a). Unify “electrophysiology FC” and “electrophysiological FC”
- b). Change “electrophysiology basis” to “electrophysiological basis”.
- c). Remove the comma (,) from “Logothetis et al., showed”
- d). The statement “our present work and a series of recent studies have consistently shown that white matter carries both functional signals and the synchronization of these signals” can be better rephrased as “our present work and a series of recent studies have consistently shown that white matter carries tract-specific, synchronized functional signals”.
- e). Change “neurobiology mechanism” to “neurobiological mechanism”.
- f). Change “we expected this result should be reproduced in a wide range of samples” to “we expect this result to be reproducible in a wide range of samples”.

g). Change “covering a small part of the entire brain” to “covering a small set of discrete brain regions”.

We apologize for these issues and thank you for pointing them out. We have addressed these points and edited the language and expressions throughout the paper together with a native speaker.

a, b) We have used ‘electrophysiological FC’ throughout the paper now (Page 13, Para 4; Page 14, Para. 1).

Our results demonstrated that white matter BOLD FC was related to white matter electrophysiological FC, which was calculated using intracranial SEEG recordings.

This association was significant across a wide range of frequency bands and for every participant, which robustly suggested the electrophysiological basis of BOLD FC in the white matter.

c) We have deleted the comma now (Page 14, Para. 1)

Logothetis et al. showed that the BOLD fMRI signal in the gray matter reflected the underlying LFPs⁴⁰, which laid the foundation of BOLD fMRI-based neuroscience studies.

d) Thank you, we have rephrased this statement (Page 14, Para. 2).

White matter has been ignored in functional brain studies for decades; however, our present work and a series of recent studies have consistently shown that white matter carries tract-specific, synchronized functional signals⁷.

e) Thank you, we have corrected it (Page 14, Para. 2).

..., which could serve as the underlying neurobiological mechanism of white matter functional signals.

f) We have edited it (Page 15, Para. 2).

Therefore, we expect this result to be reproducible in a wide range of samples.

g) We have rephrased it (Page 15, Para. 2).

Second, the SEEG recordings, which were clinically determined, were sparsely distributed in the white matter with a limited number of contacts, therefore only covering a small set of discrete brain regions.

To sum it up, the work presented is highly novel and of significant interest to the research community. There are needs for relevant descriptive details of the experimental settings and more in-depth analysis of the findings reported, in addition to thorough language improvements.

Thank you for the positive appraisal and your thoughtful comments. In the revised manuscript, we have added a detailed description of SEEG contact locations. We evaluated the correlation between BOLD FC and SEEG FC after regressing out the

distance from both matrices throughout the manuscript. Finally, the paper was edited by a native English speaker to improve the language throughout the paper.

Reviewer #2 (Remarks to the Author):

Huang and colleagues demonstrate that white matter functional connectivity, estimated by sEEG at multiple frequency bands, is correlated with the BOLD functional connectivity estimated through resting-state fMRI. They additionally demonstrate that structural connectivity estimated by diffusion tractography is significantly correlated with both BOLD and sEEG functional connectivity. The authors argue that these correlations imply an electrophysiological basis for white matter BOLD signal, and that this electrophysiology is constrained by the structural connectivity.

We thank the reviewer for the summary of our results and are grateful for the extremely useful feedback. As described below, we have incorporated all the comments into the revised manuscript.

Methods

Subjects

They included 16 patients with drug resistant epilepsy who had stereo EEG implantation - It is not clear where in the white matter exactly these electrodes were located, should include a general heatmap that shows the sampling of these regions.

This is a great suggestion. In the revised manuscript, we have provided an illustrative figure showing the electrodes of one participant (sub01) and the aggregation of all participants' electrodes in the new **Fig. S1**. We found that the contacts were mainly localized in the temporal, frontal, and parieto-temporal areas. To localize the precise anatomical location of these contacts, we used the a priori White Matter Parcellation Map (WMPM) atlas (Mori *et al.*, 2008; Oishi *et al.*, 2008), which includes 68 regions with 50 core and 18 peripheral white matter areas. We have added the precise localization of the contacts in both the main results and Supplementary Materials.

Main Results (Page 5, Para. 2):

Using an a priori White Matter Parcellation Map (WMPM) atlas^{36,37}, we found that white matter contacts were mainly localized in the temporal, frontal, and parieto-temporal areas (**Fig. S1a** and **Fig. S1b**, See **Supplementary Materials** for details).

Supplementary Materials (Page 1, Para. 1):

To localize the precise anatomical locations of the white matter contacts, we used the a priori White Matter Parcellation Map (WMPM) atlas^{1,2}, which includes 68 regions with 50 core and 18 peripheral white matter areas. First, we depicted a single participant (sub1, **Fig. S1a**) in the volume space. For better visualization, we projected all electrodes onto the same slice. We found a total of 47 white matter contacts for this participant, which were mainly localized in

the temporal blade (28 contacts), inferior frontal blade (7 contacts), and parieto-temporal blade (5 contacts) (**Fig. S1a**). Next, we aggregated all 604 white matter contacts from all participants (**Fig. S1b**). Using the WMPM atlas, we found that the contacts were mainly localized in the temporal blade (288 contacts), frontal blade (122 contacts), parieto-temporal blade (54 contacts), anterior corona radiata (21 contacts), sagittal stratum (20 contacts), posterior thalamic radiation (15 contacts).

Fig. S1 Distribution of the contacts of SEEG implanted electrodes. a) The electrodes for one participant (sub1) in volume space. To facilitate visualization, we projected all electrodes to one slice of the brain. **b)** The aggregation of all participants' electrodes visualized in surface space. Red color indicates contacts in gray matter and green color indicates contacts in white matter. There are 604 white matter contacts in total. SEEG: stereotactic EEG.

Electrode Co-Registration

The authors used an in-house co-registration pipeline. It is not clear, however, whether the final analyses, both in the sEEG and the fMRI were done in native space or MNI space. This has to be specified given it is known that registration to MNI space during the coregistration process can cause stereotactic electrodes to bend, leading to imprecise localization of the electrodes.

Thank you for pointing this out. We performed the analyses in MNI space. For clarity, we have added this point to the methods section of the revised manuscript (Page 19, Para. 1).

Next, we normalized the T1 images to MNI space and applied the acquired warping transformation to register the native coordinates of the contacts to MNI space.

We agree that the registration to MNI space could lead to imprecise electrode

localization. To validate our results, we evaluated the association between BOLD FC and SEEG FC in native space. We kept all data processing procedures except the normalization. We found that the median correlation between BOLD and SEEG white matter FC across all participants was above $r = 0.17$ in each frequency band (1–4 Hz: median $r = 0.17$; 4–8 Hz: median $r = 0.22$; 8–13 Hz: median $r = 0.23$; 13–30 Hz: median $r = 0.30$; 30–40 Hz: median $r = 0.27$; 40–70 Hz: median $r = 0.26$; 70–170 Hz: median $r = 0.22$. **Fig. S4a**). We also provide the exact r and p_{FDR} values for all participants in each frequency band in **Table S4**. Of all the 112 correlations for all participants and all frequency bands, 92 were significant ($p_{FDR} < 0.05$) after FDR correction (**Table S4**). The results are similar to our main results in the MNI space. We have added these results to the sensitivity analysis part of both the main results and Supplementary Materials.

Main Results (Page 9, Para. 2):

Briefly, we demonstrated that our results were robust to the variation of parameters in fMRI processing, including analyzing data in native space (**Fig. S4a** and **Table S4**) rather than the standard space, ...

Supplementary Materials (Page 2, Para. 2):

Here, we evaluated how fMRI preprocessing parameters impact the association between SEEG FC and BOLD FC. In our main analysis, we registered the coordinates of the SEEG contacts to the MNI space, which could lead to imprecise localization of the contacts. Here, we tested whether the results were consistent when examining the association between BOLD FC and SEEG FC in native space. We found that the median correlations between BOLD and SEEG white matter FC across all participants were higher than $r = 0.17$ in each frequency band (1–4 Hz: median $r = 0.17$; 4–8 Hz: median $r = 0.22$; 8–13 Hz: median $r = 0.23$; 13–30 Hz: median $r = 0.30$; 30–40 Hz: median $r = 0.27$; 40–70 Hz: median $r = 0.26$; 70–170 Hz: median $r = 0.22$; **Fig. S4a**). Of all the 112 correlations for all participants and all frequency bands, 92 were significant ($p_{FDR} < 0.05$) after FDR correction (**Table S4**). This result was similar to our main result.

Fig. S4a Evaluating the FC at native space, the Spearman's rank correlations between the BOLD and SEEG white matter FC were similar to the main results. Each dot represents one

participant. See **Table S4** for r and p_{FDR} of the correlations for all participants at each frequency band.

sEEG Processing

The authors broke down the sEEG signal into small time segments and then computed the Pearson correlation between the smaller segments to improve the SNR. This is a valid approach. They also ensured that no ictal activity was present in the recordings, which eliminates ictal activity as a confound from the findings. They computed thir sEEG functional connectivity across a wide range of frequency bands ranging from 1-170Hz.

Thank you for the recognition of our sEEG processing.

fMRI Preprocessing and Functional Connectivity Estimation

The authors utilized state of the art pre-processing pipelines (namely fMRIPrep and XCPengine), which allows for a standard and reproducible pre-processing of the data. It is not clear to me why they bandpassed their fMRI signal between 0.01-0.2, since the common range utilized, and the one demonstrated to allow for appropriate filtering by Satterthwaite et. al. 2014, is 0.01-0.08. This needs to be justified further.

Thank you for this comment. We appreciate the relevant literature provided. We agree that 0.01–0.08 Hz is a commonly used bandpass filtering range in the pre-processing of gray matter BOLD fMRI signals. However, the optimal bandpass filtering range in white matter BOLD fMRI is still an open question, which requires systematic studies. We tend to include a wider frequency range of bold data in white matter functional analyses.

According to the Nyquist-Shannon sampling theorem, the sampling signal can capture information from a continuous-time signal of frequency bands that are less than half the sampling frequency. The BOLD fMRI signal captured information of less than 0.25 Hz in white matter BOLD signals as TR is 2 s. Additionally, Zuo et al. have found that the slow-3 (0.073–0.198 Hz) and slow-2 (0.198–0.25 Hz) oscillations were primarily restricted to white matter compared to gray matter (see the figure below), suggesting that there could be robust activity in the white matter BOLD signal from 0.073–0.25 Hz (Zuo et al., 2010).

Figure from Zuo et al., 2010, *NeuroImage* (Zuo et al., 2010). The percentage of voxels with significant amplitude measure (i.e., fALFF) was calculated for four frequency bands: slow-5 (0.01–0.027 Hz), slow-4 (0.027–0.073 Hz), slow-3 (0.073–0.198 Hz), slow-2 (0.198–0.25 Hz). In the frequency range of slow-3 to slow-2, a high number of voxels represented significant amplitudes in the white matter.

Considering that the normal respiration rate of an adult at rest is 12 to 20 breaths per minute (0.2–0.33 Hz), we assume that signals > 0.2 Hz are respiratory or other physiological noise. Thus, we used a final frequency range of 0.01–0.20 Hz. Several prior studies have also used a higher frequency range than 0.01–0.08 Hz in white matter BOLD fMRI. For example, Peer et al., 2017, *Journal of Neuroscience* (Peer et al., 2017); Jiang et al., 2019, *NeuroImage* (Jiang et al., 2019); and Wang et al., 2022, *Cerebral Cortex* (Wang et al., 2022) applied a 0.01–0.15 Hz band-pass filtering for white matter BOLD fMRI analysis.

However, we acknowledge that there is not a gold standard for bandpass filtering in white matter BOLD fMRI processing. To avoid a bias in our results to the specific frequency range of 0.01–0.2 Hz, we re-analyzed the data in the frequency range of 0.01–0.08 Hz as recommended. We found that the results were highly consistent. Specifically, the median correlation between SEEG FC and BOLD FC across all participants was above $r = 0.16$ in each frequency band (1–4 Hz: median $r = 0.16$; 4–8 Hz: median $r = 0.20$; 8–13 Hz: median $r = 0.20$; 13–30 Hz: median $r = 0.26$; 30–40 Hz: median $r = 0.27$; 40–70 Hz: median $r = 0.24$; 70–170 Hz: median $r = 0.19$. **Fig. S4b**). We also provided the exact r and p_{FDR} values for all participants in each frequency band in **Table S5**. Of all the 112 correlations for all participants and all frequency bands, 97 were significant ($p_{FDR} < 0.05$) after FDR correction (**Table S5**). We have added this result to the sensitivity analysis in both the main results and Supplementary Materials of the revised manuscript.

Main Results (Page 9, Para. 1):

Briefly, we demonstrated that our results were robust to the variation of parameters in fMRI processing, including..., using a bandpass filtering range of 0.01–0.08 Hz (**Fig. S4b** and **Table S5**) rather than 0.01–0.2 Hz, ...

Supplementary Materials (Page 3, Para. 3):

Second, we used a bandpass filtering range of 0.01–0.2 Hz in the main results to include a wider frequency range of bold data in the white matter. Here, we evaluated a filtering range of 0.01–0.08 Hz, which is a commonly used range in the pre-processing of gray matter BOLD signals⁵. We found that the median correlations between BOLD and SEEG white matter FC across all participants were higher than $r = 0.16$ in each frequency band (1–4 Hz: median $r = 0.16$; 4–8 Hz: median $r = 0.20$; 8–13 Hz: median $r = 0.20$; 13–30 Hz: median $r = 0.26$; 30–40 Hz: median $r = 0.27$; 40–70 Hz: median $r = 0.24$; 70–170 Hz: median $r = 0.19$; **Fig. S4b**). Of all the 112 correlations for all participants and all frequency bands, 97 were significant ($p_{FDR} < 0.05$) after FDR correction (**Table S5**). This result was similar to our main result.

Fig. S4b With BOLD signals filtered using a frequency range of 0.01-0.08 Hz, the Spearman's rank correlations between the BOLD and SEEG white matter FC were similar to the main results. See **Table S5** for r and p_{FDR} of the correlations for all participants at each frequency band.

Structural Imaging Processing

The structural imaging processing was appropriate.

Thank you for the recognition of our structural imaging processing.

Main Results - Major Concerns

The findings demonstrate that the white matter BOLD and sEEG signals are significantly correlated across the selected frequency bands. The authors report that this is the case for many of the subjects at both an individual subject level and across frequency bands. However, the authors do not report the specific p-values for each subject, and it does not appear that they performed any correction for multiple comparisons across the manuscript. In the simple case of a Bonferroni procedure, there were 16 patients and 7 frequency bands, which would require a significant p-value of 0.0004 to imply significance, but the authors report at most $p < 0.001$ (more conservative approaches such as a FWER correction could also be applied). Multiple comparison correction is omitted throughout the entirety of the manuscript, and the

lack of transparency on the p-values reported makes the results difficult to interpret.

Thank you for pointing out this important issue. We now applied FDR correction for multiple comparison corrections throughout the paper. We have provided both r and FDR corrected p_{FDR} values of the correlation between BOLD and SEEG FC in the Supplementary Materials (**Table S2**). Of all the 112 correlations for all participants and all frequency bands, 104 were significant ($p_{FDR} < 0.05$) after FDR correction (**Table S2**). We also provided both r and FDR corrected p_{FDR} values of the correlation between structural connectivity and BOLD FC (**Table S12**), in which 14 of the 16 participants were significant ($p_{FDR} < 0.05$). Finally, we provided both r and FDR corrected p_{FDR} values of the correlations between structural connectivity and SEEG FC (**Table S13**). Of all the 112 correlations for all participants and all frequency bands, 95 were significant ($p_{FDR} < 0.05$) after FDR correction (**Table S13**). Finally, we also provided both r and FDR corrected p_{FDR} values for all sensitivity analyses. We have added the FDR corrected p_{FDR} values in both main results and Supplementary Materials in the revised manuscript.

Main Results (Page 8, Para. 1):

Evaluating the BOLD-SEEG correlation in white matter FC at both individual participant and individual frequency band levels, we observed that the correlations were significant with false discovery rate (FDR) corrected $p_{FDR} < 0.05$ in all seven frequency bands in 13 participants (See **Table S2** for r and p_{FDR} of the correlations for all participants at each frequency band). In the remaining three participants, the correlations were significant with $p_{FDR} < 0.05$ in three, four, and six frequency bands, respectively. Notably, the FDR correction was used to account for the multiple comparison correction across all the 16 participants and all frequency bands.

Main Results (Page 11, Para. 1):

Using FDR correction across all participants, we found 14 participants showed significant ($p_{FDR} < 0.05$) correlations between structural connectivity and BOLD white matter FC, while the other 2 participants showed no significant correlation (See **Table S12** for r and p_{FDR} for all participants).

Main Results (Page 11, Para. 2):

Using FDR correction across all participants and all frequency bands, we observed that the correlations were significant with $p_{FDR} < 0.05$ in all seven frequency bands for 8 participants (See **Table S13** for r and p_{FDR} of the correlations for all participants at each frequency band). For the other 7 participants, the correlations were significant ($p_{FDR} < 0.05$) in six frequency bands for 4 participants and were significant in five frequency bands for 3 participants. The remaining one participant (sub08) showed no significant correlation.

Table S2 The r and p_{FDR} values for the correlations between BOLD and SEEG FC for all seven different frequency bands and all 16 participants. The distance was regressed out from both FC matrices before evaluating the Spearman's rank correlation between them. Non-significant (threshold: $p_{FDR} = 0.05$) correlations were labeled with red color. BOLD: blood-oxygenation-level-dependent; SEEG: stereotactic EEG; FC: functional connectivity.

ID	r/p_{FDR}	1-4Hz	4-8Hz	8-13Hz	13-30Hz	30-40Hz	40-70Hz	70-170Hz
sub01	r	0.32	0.13	0.17	0.32	0.37	0.32	0.33
	p	1.2e-25	4.1e-05	1.7e-08	2.7e-25	6.5e-35	2.7e-25	1.2e-27
sub02	r	0.15	0.25	0.30	0.32	0.39	0.35	0.40
	p	2.3e-03	2.7e-07	3.7e-10	1.1e-11	1.1e-16	1.4e-13	2.4e-17
sub03	r	0.08	0.20	0.14	0.07	0.04	-0.01	0.00
	p	2.1e-02	9.7e-09	5.9e-05	4.9e-02	2.8e-01	7.7e-01	8.9e-01
sub04	r	0.24	0.20	0.15	0.31	0.35	0.29	0.27
	p	2.0e-09	4.3e-07	1.7e-04	4.5e-15	1.8e-18	1.9e-13	2.9e-11
sub05	r	0.21	0.23	0.24	0.26	0.31	0.32	0.30
	p	5.8e-06	1.0e-06	4.2e-07	1.5e-08	4.8e-11	1.0e-11	1.9e-10
sub06	r	0.10	0.10	0.28	0.30	0.24	0.12	0.10
	p	1.3e-01	1.2e-01	9.9e-06	1.9e-06	1.1e-04	5.1e-02	1.3e-01
sub07	r	0.12	0.10	0.11	0.13	0.13	0.14	0.08
	p	2.1e-09	2.7e-07	2.1e-08	3.1e-10	9.1e-11	8.3e-12	5.9e-05
sub08	r	0.24	0.24	0.13	0.10	0.27	0.14	0.20
	p	2.9e-08	2.4e-08	2.4e-03	1.9e-02	2.1e-10	1.6e-03	2.6e-06
sub09	r	0.20	0.32	0.37	0.46	0.40	0.36	0.32
	p	1.7e-04	1.3e-09	2.7e-12	8.4e-19	4.9e-14	1.1e-11	2.0e-09
sub10	r	0.32	0.29	0.35	0.35	0.35	0.33	0.31
	p	1.8e-08	4.2e-07	1.5e-09	7.2e-10	8.3e-10	8.2e-09	5.0e-08
sub11	r	0.16	0.19	0.17	0.19	0.20	0.17	0.18
	p	1.4e-11	1.8e-16	7.3e-13	2.4e-17	7.0e-18	4.9e-14	2.3e-14
sub12	r	0.25	0.24	0.19	0.32	0.32	0.28	0.23
	p	7.3e-17	8.8e-16	7.6e-11	5.7e-28	7.6e-28	1.7e-21	2.9e-15
sub13	r	0.14	0.11	0.12	0.23	0.31	0.28	0.29
	p	2.0e-04	3.2e-03	1.2e-03	7.2e-10	3.0e-16	1.2e-13	7.2e-15
sub14	r	0.22	0.33	0.41	0.52	0.55	0.51	0.54
	p	3.4e-04	5.0e-08	3.8e-12	8.2e-20	1.7e-22	1.6e-18	1.4e-21
sub15	r	0.17	0.28	0.33	0.22	0.16	0.15	0.12
	p	1.6e-02	6.9e-05	2.0e-06	1.3e-03	1.9e-02	2.8e-02	9.0e-02
sub16	r	0.17	0.28	0.32	0.32	0.28	0.24	0.16
	p	2.0e-02	1.2e-04	1.3e-05	1.3e-05	1.6e-04	1.1e-03	2.9e-02

Table S12 The r and p_{FDR} values for the correlations between BOLD FC and structural connectivity. The distance was regressed out from both FC and structural connectivity before evaluating the Spearman's rank correlation between them. Non-significant (threshold: $p_{FDR} = 0.05$) correlations were labeled with red color. BOLD: blood-oxygenation-level-dependent; FC: functional connectivity.

ID	sub01	sub02	sub03	sub04	sub05	sub06	sub07	sub08
r	0.16	0.39	0.40	0.39	0.22	0.27	0.33	0.12

p_{FDR}	4.8e-04	8.7e-05	1.7e-09	1.3e-06	1.5e-03	6.7e-03	9.6e-24	2.7e-01
ID	sub09	sub10	sub11	sub12	sub13	sub14	sub15	sub16
r	0.44	0.62	0.17	0.49	0.27	0.33	0.25	0.05
p_{FDR}	4.3e-06	1.8e-07	1.7e-04	7.3e-08	1.5e-02	3.9e-03	4.3e-02	5.6e-01

Table S13 The r and p_{FDR} values for the correlations between SEEG FC and structural connectivity. The distance was regressed out from both FC and structural connectivity before evaluating the Spearman's rank correlation between them. Non-significant (threshold: $p_{FDR} = 0.05$) correlations were labeled with red color. SEEG: stereotactic EEG; FC: functional connectivity.

ID	r/p_{FDR}	1-4Hz	4-8Hz	8-13Hz	13-30Hz	30-40Hz	40-70Hz	70-170Hz
sub01	r	0.36	0.23	0.35	0.33	0.29	0.29	0.29
	p	5.6e-15	4.8e-07	5.6e-15	1.5e-13	2.0e-10	3.3e-10	3.7e-10
sub02	r	0.18	0.31	0.32	0.46	0.34	0.37	0.31
	p	8.1e-02	2.6e-03	1.9e-03	3.6e-06	6.9e-04	2.4e-04	2.2e-03
sub03	r	0.21	0.23	0.35	0.35	0.39	0.35	0.32
	p	2.2e-03	6.8e-04	1.7e-07	2.3e-07	7.0e-09	2.3e-07	1.6e-06
sub04	r	0.27	0.13	0.28	0.23	0.21	0.19	0.20
	p	8.8e-04	1.0e-01	5.7e-04	5.0e-03	9.3e-03	1.8e-02	1.2e-02
sub05	r	0.10	0.18	0.30	0.31	0.26	0.29	0.22
	p	1.8e-01	9.7e-03	1.3e-05	1.2e-05	2.8e-04	3.4e-05	1.5e-03
sub06	r	0.29	0.29	0.46	0.53	0.59	0.50	0.27
	p	4.1e-03	3.9e-03	2.6e-06	2.4e-08	3.3e-10	2.8e-07	7.4e-03
sub07	r	-0.04	-0.01	0.09	0.16	0.10	0.10	0.12
	p	3.1e-01	6.9e-01	1.1e-02	2.0e-06	2.5e-03	2.6e-03	4.7e-04
sub08	r	0.14	0.05	-0.05	0.04	-0.01	-0.04	-0.05
	p	2.2e-01	6.7e-01	6.6e-01	7.3e-01	9.1e-01	7.0e-01	6.6e-01
sub09	r	0.21	0.23	0.33	0.37	0.34	0.32	0.31
	p	3.5e-02	2.0e-02	6.4e-04	1.4e-04	4.7e-04	9.1e-04	1.6e-03
sub10	r	0.44	0.20	0.16	0.33	0.47	0.49	0.47
	p	5.3e-04	1.3e-01	2.2e-01	9.7e-03	2.5e-04	1.1e-04	2.3e-04
sub11	r	0.24	0.25	0.20	0.28	0.26	0.26	0.28
	p	2.3e-07	4.0e-08	1.2e-05	8.3e-10	1.8e-08	1.8e-08	9.6e-10
sub12	r	0.13	0.28	0.32	0.35	0.42	0.42	0.38
	p	1.8e-01	2.7e-03	7.5e-04	1.8e-04	4.7e-06	6.6e-06	4.1e-05
sub13	r	0.28	0.44	0.45	0.37	0.28	0.33	0.31
	p	1.0e-02	4.8e-05	2.6e-05	6.4e-04	9.7e-03	2.6e-03	4.0e-03
sub14	r	0.18	0.13	0.27	0.40	0.37	0.35	0.33
	p	1.3e-01	2.7e-01	2.0e-02	5.7e-04	1.1e-03	2.6e-03	4.1e-03
sub15	r	0.75	0.63	0.70	0.81	0.74	0.80	0.79

	p	3.6e-12	5.7e-08	3.3e-10	5.6e-15	4.9e-12	5.6e-15	1.5e-14
sub16	r	0.44	0.43	0.25	0.28	0.31	0.27	0.33
	p	5.3e-07	1.6e-06	5.8e-03	2.2e-03	5.7e-04	2.8e-03	3.2e-04

The main finding of this study is a "statistically significant" correlation between white matter sEEG FC and BOLD FC. Additionally, there is also a correlation between a structural connectome of the same white matter regions and the sEEG and BOLD FC. The authors argue that this correlation implies an electrophysiological basis for the white matter BOLD signal. While this is potentially true, the results of the study, and the experimental design, do not answer this question. One of the studies that the authors reference is the one by Betzel et. al. 2019, where a correlation was found between cortical gray matter nodes in sEEG and BOLD. However, they seem to ignore the rest of the manuscript, where Betzel et. al. demonstrate that distance between regions has a very large influence in this correlation. In fact, one of the supplementary figures in that manuscript (supplementary figure 4), demonstrates the significant impact of Euclidean distance between regions on the correlation between the sEEG functional connectivity and other metrics. Therefore, these results are very likely to be mostly driven by a distance effect between the white matter regions, and not truly by electrophysiological signals or structural tractography, or at least that is my intuition without seeing an analysis being done to convince me otherwise. A good approach to test this would be to regress the Euclidean distance between the white matter regions and assess if the sEEG FC still predicts a significant amount of variance in the BOLD FC.

This is a great point. We agree that the distance can have an impact on the FC. To evaluate the impact of distance, we regressed out the Euclidean distance between ROIs from both matrices before examining the correlation between SEEG and BOLD white matter FC. Specifically, we computed the Euclidean distance between two contacts as $\text{distance} = \sqrt{(x_1 - y_1)^2 + (x_2 - y_2)^2 + (x_3 - y_3)^2}$, where (x_1, x_2, x_3) and (y_1, y_2, y_3) are the coordinates of the centers of two contacts. Thus, we acquired a Euclidean distance matrix for each participant, with each element in the matrix quantifying the Euclidean distance between a pair of contacts. We first regressed out the Euclidean distance from both SEEG and BOLD white matter FC and then evaluated the Spearman's rank correlation between the two residual matrices. We found that the median correlation between SEEG FC and BOLD FC across all participants was above $r = 0.19$ in each frequency band (1–4 Hz: median $r = 0.19$; 4–8 Hz: median $r = 0.23$; 8–13 Hz: median $r = 0.21$; 13–30 Hz: median $r = 0.31$; 30–40 Hz: median $r = 0.31$; 40–70 Hz: median $r = 0.28$; 70–170 Hz: median $r = 0.25$. **New Fig. 2**). Of all the 112 correlations for all participants and all frequency bands, 104 were significant ($p_{FDR} < 0.05$) after FDR correction (**Table S2**). The results suggest that the Euclidean distance between white matter contacts had a limited impact on the correlation between SEEG and BOLD FC. We have changed all the analyses in our

revised manuscript to the results controlling for distance in both main results (See new **Fig. 2**; Page 8, Para 1) and sensitivity analyses.

By repeating the above procedure, we found that for each of the other 15 participants, BOLD and SEEG white matter FC were mostly significantly correlated across all regional pairs in all frequency bands after regressing out Euclidean distance from both FC matrices (**Fig. 2**). Overall, the median Spearman's rank correlation between BOLD and SEEG white matter FC across all participants was above $r = 0.19$ in all frequency bands (1–4 Hz: median $r = 0.19$; 4–8 Hz: median $r = 0.23$; 8–13 Hz: median $r = 0.21$; 13–30 Hz: median $r = 0.31$; 30–40 Hz: median $r = 0.31$; 40–70 Hz: median $r = 0.28$; 70–170 Hz: median $r = 0.25$. **Fig. 2** and **Table S2**). Evaluating the BOLD-SEEG correlation in white matter FC at both individual participant and individual frequency band levels, we observed that the correlations were significant with false discovery rate (FDR) corrected $p_{FDR} < 0.05$ in all seven frequency bands in 13 participants (See **Table S2** for r and p_{FDR} of the correlations for all participants at each frequency band). In the remaining three participants, the correlations were significant with $p_{FDR} < 0.05$ in three, four, and six frequency bands, respectively.

New Fig. 2 The correlations between BOLD and SEEG white matter FC in all frequency bands for all the 16 participants. The Euclidean distances between pairs of regions were regressed out from both BOLD and SEEG FC before evaluating the correlations. There are 16 dots in each frequency band, representing the participants. The median correlations between BOLD and SEEG white matter FC across all participants were higher than $r = 0.19$ in all frequency bands (1-4 Hz: median $r = 0.19$; 4-8 Hz: median $r = 0.23$; 8-13 Hz: median $r = 0.21$; 13-30 Hz: median $r = 0.31$; 30-40 Hz: median $r = 0.31$; 40-70 Hz: median $r = 0.28$; 70-170 Hz: median $r = 0.25$). The correlations were significant with $p_{FDR} < 0.05$ in all seven frequency bands for 13 participants and the remaining three participants exhibited significant ($p_{FDR} < 0.05$) correlations in three, four, and six frequency bands, respectively. See **Table S2** for r and p_{FDR} of the correlations for all participants at each frequency band. BOLD: blood-oxygenation-level-dependent; SEEG: stereotactic EEG; FC: functional connectivity.

Similarly, we regressed out the distance from both the structural connectivity and white matter FC before examining the Spearman's rank correlation between them, and the results reduced compared to our original results but were still highly significant. We found that the median Spearman's rank correlation between structural connectivity and BOLD FC was $r = 0.30$ (see **Table S12** for r and p_{FDR} of all participants). Using FDR

correction, we found 14 of the 16 participants showed significant ($p_{FDR} < 0.05$) correlations. We also found that the median correlation between structural connectivity and SEEG FC across all participants was above $r = 0.22$ in each frequency band (1–4 Hz: median $r = 0.22$; 4–8 Hz: median $r = 0.23$; 8–13 Hz: median $r = 0.31$; 13–30 Hz: median $r = 0.34$; 30–40 Hz: median $r = 0.33$; 40–70 Hz: median $r = 0.33$; 70–170 Hz: median $r = 0.31$. see New **Fig. 3f**). Of all the 112 correlations for all participants and all frequency bands, 95 were significant ($p_{FDR} < 0.05$) after FDR correction (**Table S13**). The results suggest that the Euclidean distance between white matter contacts had only a limited impact on the correlation between SEEG FC and structural connectivity. We have changed all the analyses in our revised manuscript to the results controlling for distance (See new **Fig. 3f**; Page 11, Para 1 and 2).

Finally, our results showed that the structural connectivity correlated with BOLD white matter FC in all participants after regressing out the distance (median $r = 0.30$, **Fig. 3c**). Using FDR correction across all participants, we found 14 participants showed significant ($p_{FDR} < 0.05$) correlations between structural connectivity and BOLD white matter FC, while the other 2 participants showed no significant correlation (See **Table S12** for r and p_{FDR} for all participants).

We next evaluated the coupling between structural connectivity and SEEG white matter FC after regressing out the distance from both matrices. Similar to BOLD FC, the SEEG white matter FC (1–4 Hz) was also significantly correlated ($r = 0.36$, $p_{FDR} < 0.001$) with structural connectivity across regional pairs with non-zero structural connections (sub1, **Fig. 3d and 3e**). We also found that the median Spearman’s rank correlation between SEEG white matter FC and structural connectivity across all participants was above $r = 0.22$ in each frequency band (1–4 Hz: median $r = 0.22$; 4–8 Hz: median $r = 0.23$; 8–13 Hz: median $r = 0.31$; 13–30 Hz: median $r = 0.34$; 30–40 Hz: median $r = 0.33$; 40–70 Hz: median $r = 0.33$; 70–170 Hz: median $r = 0.31$; **Fig. 3f**). Using FDR correction across all participants and all frequency bands, we observed that the correlations were significant with $p_{FDR} < 0.05$ in all seven frequency bands for 8 participants (See **Table S13** for r and p_{FDR} of the correlations for all participants at each frequency band). For the other 7 participants, the correlations were significant ($p_{FDR} < 0.05$) in six frequency bands for 4 participants and were significant in five frequency bands for 3 participants. The remaining one participant (sub08) showed no significant correlation.

New Fig. 3f The median correlations between structural connectivity and SEEG white matter FC across all participants were higher than $r = 0.22$ in each frequency band (1-4 Hz: median $r = 0.22$; 4-8 Hz: median $r = 0.23$; 8-13 Hz: median $r = 0.31$; 13-30Hz: median $r = 0.34$; 30-40 Hz: median $r = 0.33$; 40-70 Hz: median $r = 0.33$; 70-170 Hz: median $r = 0.31$). See **Table S13** for r and p_{FDR} of the correlations for all participants at each frequency band. Notably, the Euclidean distances between pairs of regions were regressed out from structural connectivity, SEEG FC and BOLD FC before evaluating the correlations between matrices.

Because the purpose of this study is to demonstrate a "novel" correlation in the white matter between sEEG and BOLD, there has to be significantly more rigor in testing what is driving this correlation. For example, there should be appropriate null models to compare the correlations (e.g. is the correlation between WM sEEG FC and BOLD FC higher than that of WM sEEG FC and the Euclidean distance matrix between the electrodes?).

We appreciate your suggestion. As described above, we have computed the Euclidean distance between the centers of each pair of contacts. We have computed the correlation between the SEEG FC matrix and the matrix of Euclidean distance across all edges. We found that the median correlation between SEEG FC and distance across all participants ranged from $r = -0.72$ to $r = -0.48$ across all frequency bands (1–4 Hz: median $r = -0.48$; 4–8 Hz: median $r = -0.58$; 8–13 Hz: median $r = -0.61$; 13–30 Hz: median $r = -0.59$; 30–40 Hz: median $r = -0.60$; 40–70 Hz: median $r = -0.72$; 70–170 Hz: median $r = -0.71$. **Fig. S3**). We provide the exact r and p_{FDR} values for all participants in each frequency band in **Table S3**. These results suggested the FC reduced with longer distance between contacts, which was consistent with prior literature (Betzel *et al.*, 2019). Regarding the effect size (absolute correlation coefficient), our association between SEEG and BOLD FC were lower than the association between SEEG FC and Euclidean distance. It should be noted that the correlation between SEEG FC and Euclidean distance were similar to prior literatures. For example, in figure 4a of Betzel *et al.*, 2019, Nat. Biomed. Eng., they showed the ECoG FC was significantly associated with Euclidean distance with a prominent significance ($P < 10^{-15}$) (Betzel *et al.*, 2019). In contrast to SEEG FC, BOLD FC exhibited a lower correlation (median $r = -0.3$ across all participants) with distance, which was similar to the effect size reported in prior literature (Mišić *et al.*, 2014). Thus, distance could have a limited impact on the correlation between SEEG FC and BOLD FC. Finally, as our response to the comments above, the correlations between SEEG and BOLD white matter FC were still significant after regressing out the distance from both matrices, and now we regressed out the distance throughout the manuscript in both main results and the sensitivity analyses. We have added the correlations between SEEG FC and distance as well as the correlations between BOLD FC and distance to the Supplementary Materials (Page 2, Para 2) of revised manuscript to justify controlling the Euclidean distance in the association between SEEG and BOLD white matter FC.

Prior studies have reported an association between distance and FC^{3,4}; here, we evaluated the association between SEEG FC and Euclidean distance between contacts. We computed the

Euclidean distance between each pair of contacts using the following formula: distance = $\sqrt{(x_1 - y_1)^2 + (x_2 - y_2)^2 + (x_3 - y_3)^2}$, where (x_1, x_2, x_3) and (y_1, y_2, y_3) are the coordinates of the centers of the two respective contacts. We constructed a distance matrix for each participant, with each element in the matrix quantifying the Euclidean distance between a pair of contacts. We found the SEEG FC was highly correlated with the distance across all participants in each frequency band. The median correlations between SEEG FC and distance ranged from $r = -0.72$ to $r = -0.48$ across all frequency bands (1–4 Hz: median $r = -0.48$; 4–8 Hz: median $r = -0.58$; 8–13 Hz: median $r = -0.61$; 13–30 Hz: median $r = -0.59$; 30–40 Hz: median $r = -0.60$; 40–70 Hz: median $r = -0.72$; 70–170 Hz: median $r = -0.71$; **Fig. S3**). See **Table S3** for r and p_{FDR} of the correlations for all participants at each frequency band. This result was similar to prior literature (See Fig. 4a in Betzel et al. 2019, Nat. Biomed. Eng.³). Similarly, BOLD FC also showed significant correlations with distance (median $r = -0.3$ across all participants), which was similar to the results reported in prior literature⁴. Therefore, we regressed out the distance from both matrices before examining the correlation between SEEG FC and BOLD FC throughout the manuscript. Similarly, as prior studies consistently reported an association between distance and structural connectivity^{3,5}, we also regressed out the distance from both matrices before evaluating the correlation between structural connectivity and other connectivity matrices. Notably, distance was regressed out in both the main analyses and all sensitivity analyses.

Fig. S3 Spearman’s rank correlation between SEEG FC and contact distance. Each dot represents one participant. See **Table S3** for r and p_{FDR} of the correlations for all participants at each frequency band. SEEG: stereotactic EEG; FC: functional connectivity.

There should also be additional sensitivity analyses: how does fMRI preprocessing affect these correlations, does the window size matter for the sEEG connectivity, does the radius of the electrode ROI affect the findings, do other correlation metrics more commonly used in sEEG FC such as coherence provide different findings?

We appreciate this comment and agree that the sensitivity analyses are critical. In the response to the comments above, we have demonstrated that our results are similar when analyzing in native space or using a filtering range of 0.01–0.08 Hz, suggesting

the robustness of our results to fMRI preprocessing parameters. We further evaluated whether our results still hold when regressing out the global and CSF signals. The results indicated that the median correlations between BOLD and SEEG white matter FC across all participants were above $r = 0.18$ in each frequency band (1–4 Hz: median $r = 0.18$; 4–8 Hz: median $r = 0.24$; 8–13 Hz: median $r = 0.24$; 13–30 Hz: median $r = 0.29$; 30–40 Hz: median $r = 0.31$; 40–70 Hz: median $r = 0.25$; 70–170 Hz: median $r = 0.22$; **Fig. S4c**). We provide the exact r and p_{FDR} values for all participants in each frequency band in **Table S6**. Of all the 112 correlations for all participants and all frequency bands, 111 were significant ($p_{FDR} < 0.05$) after FDR correction (**Table S6**). This result was very similar to our main results without global and CSF signals regression in the fMRI preprocessing. We have added this result to the sensitivity analysis in both the main results and Supplementary Materials of the revised manuscript.

Main Results (Page 9, Para. 1):

Briefly, we demonstrated that our results were robust to the variation of parameters in fMRI processing, including..., regressing out the global and CSF signals during pre-processing (**Fig. S4c** and **Table S6**), ...

Supplementary Materials (Page 3, Para. 2):

Third, we tested whether our results would be consistent after regressing out the global and CSF signals during fMRI preprocessing. We found that the median correlations between BOLD and SEEG white matter FC across all participants were higher than $r = 0.18$ in each frequency band (1–4 Hz: median $r = 0.18$; 4–8 Hz: median $r = 0.24$; 8–13 Hz: median $r = 0.24$; 13–30 Hz: median $r = 0.29$; 30–40 Hz: median $r = 0.31$; 40–70 Hz: median $r = 0.25$; 70–170 Hz: median $r = 0.22$; **Fig. S4c**). Of all the 112 correlations for all participants and all frequency bands, 111 were significant ($p_{FDR} < 0.05$) after FDR correction (**Table S6**). This result was similar to our main result.

Fig. S4c With global and CSF signals regressed out during fMRI preprocessing, the Spearman's rank correlations between BOLD and SEEG white matter FC were similar to the main results. See **Table S6** for r and p_{FDR} of the correlations for all participants at each frequency band.

Next, we evaluated whether the choice of window size of SEEG data processing

affected our results. In our main results, we used 10 consecutive segments, each comprise a windows size of 6 s; here, we tested whether our results hold with windows sizes of 4 s or 8 s. Using a window size of 4 s, we found that the median correlations between BOLD and SEEG white matter FC across all participants were above $r = 0.16$ in each frequency band (1–4 Hz: median $r = 0.16$; 4–8 Hz: median $r = 0.26$; 8–13 Hz: median $r = 0.26$; 13–30 Hz: median $r = 0.30$; 30–40 Hz: median $r = 0.33$; 40–70 Hz: median $r = 0.28$; 70–170 Hz: median $r = 0.24$. **Fig. S6a**). We provide the exact r and p_{FDR} values for all participants in each frequency band in **Table S9**. Of all the 112 correlations for all participants and all frequency bands, 100 were significant ($p_{FDR} < 0.05$) after FDR correction (**Table S9**). We also obtained consistent results with a window size of 8 s (**Fig. S6b and Table S10**). These results suggest that the choice of windows size does not affect our main results. We have added this result to the sensitivity analysis in both the main results and Supplementary Materials of the revised manuscript.

Main Results (Page 10, Para. 2):

Our results were also robust to variation in parameters during SEEG data processing. We used 10 consecutive segments of the SEEG time series data with a length of 6 s, respectively, in the main analysis. Here, we tested 10 segments with a respective length of 4 s or 8 s, and found that the results were similar to our main results (see **Fig. S6a and Table S9** for 4 s; see **Fig. S6b and Table S10** for 8 s).

Supplementary Materials (Page 4, Para. 2):

First, we evaluated whether the choice of window size for the SEEG data affected our main results. We used 10 consecutive segments of the SEEG time series data with a window size of 6 s, respectively, in our main results, and further tested a window size of 4 s and 8 s. Using a window size of 4 s, we found that the median correlations between BOLD and SEEG white matter FC across all participants were higher than $r = 0.16$ in each frequency band (1–4 Hz: median $r = 0.16$; 4–8 Hz: median $r = 0.26$; 8–13 Hz: median $r = 0.26$; 13–30 Hz: median $r = 0.30$; 30–40 Hz: median $r = 0.33$; 40–70 Hz: median $r = 0.28$; 70–170 Hz: median $r = 0.24$; **Fig. S6a**). Of all the 112 correlations for all participants and all frequency bands, 100 were significant ($p_{FDR} < 0.05$) after FDR correction (**Table S9**). This result was similar to our main result. We also found a consistent result with the 8 s window size (**Fig. S6b and Table S10**).

Fig. S6 Spearman's rank correlations between BOLD and SEEG FC were assessed using different window sizes for SEEG analysis. a) With a window size of 4 s, the correlations between BOLD and SEEG white matter FC were similar to the main results. Each dot represents one participant. See **Table S9** for r and p_{FDR} of the correlations for all participants at each frequency band. **b)** With a window size of 8 s, the results were similar to the main results. Each dot represents one participant. See **Table S10** for r and p_{FDR} of the correlations for all participants at each frequency band. BOLD: blood-oxygenation-level-dependent; SEEG: stereotactic EEG; FC: functional connectivity.

We further evaluated whether the radius of the contact ROI for the BOLD FC calculation affected our main result of the correlation between SEEG and BOLD FC. Considering the center to center distance between adjacent contacts was 3.5 mm and the resolution of one fMRI voxel was 2 mm after resampling, we tested different ROI size in a limited range. In our main analysis, we defined the contact ROI with a radius 3mm, and the implementation was to include the neighboring voxels adjacent to the surface and edge, which contained 19 voxels in total. Here, we further tested the definition of ROI comprising neighboring voxels adjacent to the surface, which included 7 voxels, as well as the definition of ROI comprising neighboring voxels adjacent to the surface, edge and vertex, which included 27 voxels. Using ROI definition of 7 voxel neighbors, we found that the median correlations between BOLD and SEEG white matter FC across all participants were above $r = 0.17$ in each frequency band (1–4 Hz: median $r = 0.17$; 4–8 Hz: median $r = 0.21$; 8–13 Hz: median $r = 0.22$; 13–30 Hz: median $r = 0.27$; 30–40 Hz: median $r = 0.29$; 40–70 Hz: median $r = 0.27$; 70–170 Hz: median $r = 0.23$; **Fig. S5a**). We provide the exact r and p_{FDR} values for all participants in each frequency band in **Table S7**. Of all the 112 correlations for all participants and all frequency bands, 101 were significant ($p_{FDR} < 0.05$) after FDR correction (**Table S7**). This result was similar to our main result. We also obtained a consistent result with ROI defined as 27 voxel neighbors (**Fig. S5b** and **Table S8**). This result suggests that the choice of radius of the ROI did not affect our main results. We have added this result to the sensitivity analysis in both the main results (Page 10, Para. 1) and Supplementary Materials (Page 3, Para. 3) of the revised manuscript.

Main Results:

Briefly, we demonstrated that our results were robust to the variation of parameters in fMRI processing, including ..., and using 7 voxels neighbors (**Fig. S5a** and **Table S7**) or 27 voxels neighbors (**Fig. S5b** and **Table S8**) to define the ROIs for the FC calculation.

Supplementary Materials:

Finally, we evaluated how the radius of the ROI for the BOLD FC calculation affected the correlation between SEEG FC and BOLD FC. In the main analysis, we defined the contact ROI comprising neighboring voxels adjacent to the surface and edge, which contained 19 voxels in total. Here, we tested the definition of ROI comprising neighboring voxels adjacent to the surface, which included 7 voxels, as well as the definition of ROI comprising neighbors adjacent to the surface, edge and vertex, which included 27 voxels. Using ROI definition of 7 voxels neighbors, we found that the median correlations between BOLD and SEEG white matter FC across all participants were higher than $r = 0.17$ in each frequency band (1–4 Hz: median $r = 0.17$; 4–8 Hz: median $r = 0.21$; 8–13 Hz: median $r = 0.22$; 13–30 Hz: median $r = 0.27$; 30–40 Hz: median $r = 0.29$; 40–70 Hz: median $r = 0.27$; 70–170 Hz: median $r = 0.23$; **Fig. S5a**). Of all the 112 correlations for all participants and all frequency bands, 101 were significant ($p_{FDR} < 0.05$) after FDR correction (**Table S7**). This result was similar to our main result. We also obtained a consistent result with ROI defined as 27 voxels neighbors (See **Fig. S5b** and **Table S8**).

Fig. S5 Spearman's rank correlations between BOLD and SEEG FC were evaluated using different ROI size for BOLD FC calculation. a) With the definition of ROI comprising 7 neighboring voxels adjacent to the surface, the correlations between BOLD and SEEG white matter FC were similar to the main results. Each dot represents one participant. See **Table S7** for r and p_{FDR} of the correlations for all participants at each frequency band. **b)** With the definition of ROI comprising 27 neighboring voxels adjacent to the surface, edge and vertex, the correlations between BOLD and SEEG white matter FC were similar to the main results. See **Table S8** for r and p_{FDR} of the correlations for all participants at each frequency band. BOLD: blood-oxygenation-level-dependent; SEEG: stereotactic EEG; FC: functional connectivity.

Finally, as suggested, we evaluated whether our results hold with coherence-based

SEEG FC. We found that the median correlations between BOLD and SEEG white matter FC across all participants were above $r = 0.22$ in each frequency band (1–4 Hz: median $r = 0.23$; 4–8 Hz: median $r = 0.22$; 8–13 Hz: median $r = 0.23$; 13–30 Hz: median $r = 0.24$; 30–40 Hz: median $r = 0.25$; 40–70 Hz: median $r = 0.27$; 70–170 Hz: median $r = 0.26$; **Fig. S7** and **Table S11**). Of all the 112 correlations for all participants and all frequency bands, 105 were significant ($p_{FDR} < 0.05$) after FDR correction (**Table S11**). This result suggests that using coherence or Pearson’s correlation to evaluate SEEG FC did not significantly impact our results. We have added this result to the sensitivity analysis in both the main results and Supplementary Materials of the revised manuscript.

Main Results (Page 10, Para. 2):

We used Pearson’s correlation to evaluate the SEEG FC in the main analysis, and here we found that coherence-based SEEG FC also exhibited similar correlations with BOLD FC (**Fig. S7** and **Table S11**).

Supplementary Materials (Page 4, Para. 3):

Second, in our main results, we evaluated the SEEG FC using the Pearson’s correlation between the time series of two contacts. Here, we evaluated the coupling between SEEG and BOLD FC for a coherence-based SEEG FC. We found that the median correlations between BOLD and SEEG white matter FC across all participants was higher than $r = 0.22$ in each frequency band (1–4 Hz: median $r = 0.23$; 4–8 Hz: median $r = 0.22$; 8–13 Hz: median $r = 0.23$; 13–30 Hz: median $r = 0.24$; 30–40 Hz: median $r = 0.25$; 40–70 Hz: median $r = 0.27$; 70–170 Hz: median $r = 0.26$; see **Fig. S7**). Of all the 112 correlations for all participants and all frequency bands, 105 were significant ($p_{FDR} < 0.05$) after FDR correction (**Table S11**). This result suggested that coherence-based SEEG FC also showed correlations with BOLD FC.

Fig. S7 Spearman’s rank correlations between BOLD FC and coherence-based SEEG FC. Each dot represents one participant. See **Table S11** for r and p_{FDR} of the correlations for all participants at each frequency band. BOLD: blood-oxygenation-level-dependent; SEEG: stereotactic EEG; FC: functional connectivity.

Overall the manuscript addresses an important problem, but lacks the scientific rigor necessary to ensure that the results support the conclusions the authors make on them,

and to ensure that the findings are robust and reproducible.

Thank you for the recognition of our work. We appreciate your comments, which helped us improve the robustness of our results. We provided a series of additional analyses to increase the robustness and reproducibility of our findings.

General Summary of Impressions

Huang and colleagues demonstrate that white matter functional connectivity, estimated by sEEG at multiple frequency bands, is correlated with the BOLD functional connectivity estimated through resting-state fMRI. They additionally demonstrate that structural connectivity estimated by diffusion tractography is significantly correlated with BOLD and sEEG functional connectivity. While the findings are novel, there are multiple serious theoretical and methodological considerations that decrease the quality and interpretation of the findings. First, this is an exploratory study, therefore care has to be taken in correcting for spurious effects and multiple comparisons. The authors did not do this throughout the manuscript. Second, the authors are trying to establish the existence of a correlation between WM sEEG FC and WM BOLD FC, therefore significantly more rigor (appropriate null models, sensitivity analyses, etc.) is required to ensure validity of the results. Specifically, it is well known that these correlation metrics are highly influenced by the distance between the nodes (either voxels, ROIs, or electrodes), therefore these results are very likely to be mostly driven by a distance effect between the white matter regions, and not truly by electrophysiological signals or structural tractography. Overall the manuscript addresses an important problem, but lacks the scientific rigor necessary to ensure that the results support the conclusions the authors make on them, and to ensure that the findings are reproducible.

We appreciate your summary and recognition of the novelty of our study and thank you for the valuable comments. As detailed above, we have addressed these comments with a series of additional analyses and have included the results in our revised manuscript. To address the problem of multiple comparisons, we used FDR correction for all analyses across the entire paper. We demonstrated that the correlation between SEEG FC and BOLD FC in the white matter was still significant after regressing out the distance between ROIs from both matrices. Similarly, the distance had a limited impact in the association between structural connectivity and both BOLD as well as between structural connectivity and SEEG FC. We have now controlled for the distance throughout all analyses in the manuscript. We also performed sensitivity analyses to demonstrate the robustness and reproducibility of our results to variations in the data processing parameters. We demonstrated that our results were stable with different fMRI preprocessing parameters, including analyzing in native space, using a filtering range of 0.01–0.08 Hz, and regressing out the global and CSF signals. We also found that our results still hold with different window sizes for SEEG connectivity and different contact ROI size for BOLD connectivity. Moreover, a different approach (coherence) to quantify the SEEG connectivity returned a similar

result to our main analyses with Pearson's correlation for connectivity. Overall, we found that our conclusions are robust to variations in data preprocessing and analyses.

Reviewer #3 (Remarks to the Author):

The authors present an important study comparing intracranial electrophysiological and BOLD fMRI data in white matter. The results provide novel evidence to further support functional MRI in white matter, an area I know well. While the study overall is commendable, there are some outstanding issues and questions to address:

We appreciate this positive appraisal from the reviewer and are grateful for the thoughtful comments.

Major:

1) It is critically important to add a caveat in the discussion that these results are in patients who have seizure severity requiring surgical intervention - and therefore the impacts of this likely affect the underlying physiological relationship relative to individuals who do not experience seizures (ie, generalizability remains to be demonstrated).

This is an important consideration. We have added this to the limitations section of the revised manuscript (Page 15, Para. 2).

Fourth, the findings reported here were evaluated in medical-resistant epilepsy patients who had seizure severity requiring surgical intervention. Whether our results could be generalized to healthy populations remains unclear.

2) Please explain why Spearman's correlations were used rather than Pearson's correlations. Is there a rationale for a non-parametric approach? It would be good to have also run Pearson's to know if this was insensitive and why?

Thank you for this comment. We used Spearman's rank correlation because FC was not normally distributed. As suggested, we calculated the correlation between SEEG and BOLD white matter FC using Pearson's correlation. Notably, according to the comments from other reviewers, we now regressed out the distance from both SEEG FC and BOLD FC matrices throughout the manuscript, which reduced the correlations compared to our original results. Using Pearson's correlation, we found the median correlations between SEEG and BOLD FC were higher than $r = 0.22$ in each frequency band (1–4 Hz: median $r = 0.22$; 4–8 Hz: median $r = 0.26$; 8–13 Hz: median $r = 0.26$; 13–30 Hz: median $r = 0.34$; 30–40 Hz: median $r = 0.36$; 40–70 Hz: median $r = 0.34$; 70–170 Hz: median $r = 0.30$; **Fig. R1** below). Using FDR correction, we found the association between SEEG FC and BOLD FC was significant ($p_{FDR} < 0.05$) for all seven frequency bands in 15 participants, and in the remaining participant the association

was significant in five frequency band (**Table R1** below). These results were slightly higher than the results using Spearman's rank correlation (See **Fig.2** and **Table S2**). As the FC was not normally distributed, we keep using the Spearman's correlation in our analyses. Now we have added this rationale in the revised manuscript (Page 6, Para. 1).

We next used Spearman's rank correlation to evaluate the similarity between the two FC matrices as the FC was not normally distributed.

Fig. R1 Correlations between BOLD and SEEG white matter FC using Pearson's correlation. Each dot represents one participant. See **Table R1** for r and p_{FDR} of the correlations for all participants at each frequency band. BOLD: blood-oxygenation-level-dependent; SEEG: stereotactic EEG; FC: functional connectivity.

Table R1 The r and p_{FDR} values for the correlations between BOLD and SEEG FC across seven different frequency bands for all 16 participants using Pearson's correlation. The distance was regressed out from both FC matrices before evaluating the correlation. Non-significant (threshold: $p_{FDR} = 0.05$) correlations were labeled with red color. SEEG: stereotactic EEG; FC: functional connectivity.

ID	r/p_{FDR}	1-4Hz	4-8Hz	8-13Hz	13-30Hz	30-40Hz	40-70Hz	70-170Hz
sub01	r	0.32	0.13	0.21	0.36	0.41	0.36	0.38
	p	4.2e-27	2.7e-05	4.8e-12	1.3e-33	3.1e-44	4.2e-34	4.5e-37
sub02	r	0.18	0.29	0.34	0.35	0.41	0.39	0.44
	p	2.0e-04	2.0e-09	1.4e-12	1.6e-13	2.1e-18	1.0e-16	6.2e-22
sub03	r	0.12	0.23	0.22	0.20	0.15	0.17	0.15
	p	3.8e-04	7.9e-12	2.9e-10	5.2e-09	1.0e-05	1.2e-06	6.8e-06
sub04	r	0.26	0.24	0.21	0.34	0.37	0.32	0.30
	p	5.8e-11	2.7e-09	1.3e-07	1.1e-17	1.8e-21	2.6e-16	5.0e-14
sub05	r	0.22	0.29	0.29	0.34	0.35	0.37	0.34
	p	1.9e-06	1.9e-10	5.6e-10	9.4e-14	3.2e-14	4.0e-16	1.6e-13
sub06	r	0.10	0.11	0.19	0.18	0.18	0.16	0.14
	p	1.1e-01	8.2e-02	2.3e-03	3.5e-03	4.5e-03	1.0e-02	2.4e-02
sub07	r	0.15	0.14	0.19	0.25	0.22	0.23	0.16
	p	9.2e-14	3.0e-12	3.4e-21	1.1e-35	6.7e-29	4.8e-29	1.2e-14

sub08	r	0.23	0.22	0.17	0.15	0.27	0.16	0.21
	p	1.0e-07	1.2e-07	4.8e-05	4.8e-04	2.3e-10	1.1e-04	4.0e-07
sub09	r	0.25	0.35	0.41	0.49	0.44	0.41	0.36
	p	2.5e-06	2.6e-11	1.4e-15	2.3e-22	9.5e-18	1.4e-15	6.1e-12
sub10	r	0.30	0.30	0.36	0.37	0.38	0.36	0.35
	p	1.3e-07	2.4e-07	3.6e-10	5.2e-11	3.1e-11	4.1e-10	4.5e-10
sub11	r	0.19	0.23	0.20	0.25	0.27	0.25	0.27
	p	2.0e-17	8.0e-25	1.3e-18	4.8e-29	8.4e-34	2.9e-29	6.0e-33
sub12	r	0.32	0.33	0.30	0.43	0.40	0.39	0.30
	p	6.1e-28	1.3e-30	8.4e-25	2.6e-52	8.5e-46	4.7e-42	3.8e-25
sub13	r	0.26	0.21	0.23	0.34	0.39	0.42	0.42
	p	8.4e-12	1.5e-08	6.4e-10	1.4e-19	4.7e-26	5.0e-31	1.8e-30
sub14	r	0.25	0.36	0.44	0.55	0.57	0.55	0.57
	p	4.7e-05	1.3e-09	5.3e-14	2.8e-22	5.8e-25	2.3e-22	1.7e-24
sub15	r	0.17	0.30	0.36	0.26	0.22	0.21	0.18
	p	1.5e-02	9.7e-06	9.9e-08	2.0e-04	1.6e-03	2.2e-03	1.1e-02
sub16	r	0.18	0.28	0.31	0.36	0.32	0.28	0.21
	p	1.5e-02	1.1e-04	1.4e-05	4.9e-07	1.1e-05	1.2e-04	4.6e-03

Minor:

The cited literature is not up-to-date. A fair amount of recent studies should be cited to summarize recent advances in both characterizing the white matter HRF and sensitivity to white matter neuroplasticity to name a few areas that demonstrate the growing validity of white matter fMRI.

Thank you for pointing this out. We have added four references on the characterization of white matter HRF (*Courtemanche et al., 2018; Li et al., 2019; Schilling et al., 2022; Wang et al., 2020*) and three references on the white matter neuroplasticity with BOLD fMRI (*Frizzell et al., 2020; Frizzell et al., 2022; Kirby et al., 2022*) to both the Introduction and Discussion sections.

Introduction (Page 3, Para. 1)

Moreover, recent studies have characterized the hemodynamic response function^{5,10,15,16} and neuroplasticity¹⁷⁻¹⁹ in white matter using BOLD fMRI.

Discussion (Page 14, Para. 2)

For example, recent studies have characterized the BOLD hemodynamic response function in the white matter, which displayed both task- and tract-specific patterns, distinct from that in the gray matter^{5,10,15,16}. It has been also shown that the functional neuroplasticity in white matter tracts caused by motor learning could be detected using BOLD fMRI¹⁷⁻¹⁹.

References:

- Betzel, R.F., Medaglia, J.D., Kahn, A.E., Soffer, J., Schonhaut, D.R., and Bassett, D.S. (2019). Structural, geometric and genetic factors predict interregional brain connectivity patterns probed by electrocorticography. *Nat Biomed Eng* 3, 902-916. 10.1038/s41551-019-0404-5.
- Courtemanche, M.J., Sparrey, C.J., Song, X., MacKay, A., and D'Arcy, R.C.N. (2018). Detecting white matter activity using conventional 3 Tesla fMRI: An evaluation of standard field strength and hemodynamic response function. *Neuroimage* 169, 145-150. 10.1016/j.neuroimage.2017.12.008.
- Frizzell, T.O., Grajauskas, L.A., Liu, C.C., Ghosh Hajra, S., Song, X., and D'Arcy, R.C.N. (2020). White Matter Neuroplasticity: Motor Learning Activates the Internal Capsule and Reduces Hemodynamic Response Variability. *Front Hum Neurosci* 14, 509258. 10.3389/fnhum.2020.509258.
- Frizzell, T.O., Phull, E., Khan, M., Song, X., Grajauskas, L.A., Gawryluk, J., and D'Arcy, R.C.N. (2022). Imaging functional neuroplasticity in human white matter tracts. *Brain Struct Funct* 227, 381-392. 10.1007/s00429-021-02407-4.
- Hansen, J.Y., Shafiei, G., Markello, R.D., Smart, K., Cox, S.M.L., Norgaard, M., Beliveau, V., Wu, Y., Gallezot, J.D., Aumont, E., et al. (2022). Mapping neurotransmitter systems to the structural and functional organization of the human neocortex. *Nat Neurosci*. 10.1038/s41593-022-01186-3.
- Iida, K., and Otsubo, H. (2017). Stereoelectroencephalography: Indication and Efficacy. *Neurol Med Chir (Tokyo)* 57, 375-385. 10.2176/nmc.ra.2017-0008.
- Jiang, Y., Luo, C., Li, X., Li, Y., Yang, H., Li, J., Chang, X., Li, H., Yang, H., Wang, J., et al. (2019). White-matter functional networks changes in patients with schizophrenia. *Neuroimage* 190, 172-181. 10.1016/j.neuroimage.2018.04.018.
- Kirby, E.D., Frizzell, T.O., Grajauskas, L.A., Song, X., Gawryluk, J.R., Lakhani, B., Boyd, L., and D'Arcy, R.C.N. (2022). Increased myelination plays a central role in white matter neuroplasticity. *Neuroimage* 263, 119644. 10.1016/j.neuroimage.2022.119644.
- Li, M., Newton, A.T., Anderson, A.W., Ding, Z., and Gore, J.C. (2019). Characterization of the hemodynamic response function in white matter tracts for event-related fMRI. *Nat Commun* 10, 1140. 10.1038/s41467-019-09076-2.
- Mišić, B., Fatima, Z., Askren, M.K., Buschkuhl, M., Churchill, N., Cimprich, B., Deldin, P.J., Jaeggi, S., Jung, M., Korostil, M., et al. (2014). The Functional Connectivity Landscape of the Human Brain. *PLoS One* 9, e111007. 10.1371/journal.pone.0111007.
- Mori, S., Oishi, K., Jiang, H., Jiang, L., Li, X., Akhter, K., Hua, K., Faria, A.V., Mahmood, A., Woods, R., et al. (2008). Stereotaxic white matter atlas based on diffusion tensor imaging in an ICBM template. *Neuroimage* 40, 570-582. 10.1016/j.neuroimage.2007.12.035.
- Oishi, K., Zilles, K., Amunts, K., Faria, A., Jiang, H., Li, X., Akhter, K., Hua, K., Woods, R., Toga, A.W., et al. (2008). Human brain white matter atlas: identification and assignment of common anatomical structures in superficial white matter. *Neuroimage* 43, 447-457. 10.1016/j.neuroimage.2008.07.009.
- Peer, M., Nitzan, M., Bick, A.S., Levin, N., and Arzy, S. (2017). Evidence for Functional Networks within the Human Brain's White Matter. *The Journal of neuroscience : the official journal of the Society for Neuroscience* 37, 6394-6407. 10.1523/JNEUROSCI.3872-16.2017.
- Schilling, K.G., Li, M., Rheault, F., Ding, Z., Anderson, A.W., Kang, H., Landman, B.A., and

Gore, J.C. (2022). Anomalous and heterogeneous characteristics of the BOLD hemodynamic response function in white matter. *Cereb Cortex Commun* 3, tgac035. 10.1093/texcom/tgac035.

Wang, P., Wang, J., Michael, A., Wang, Z., Klugah-Brown, B., Meng, C., and Biswal, B.B. (2022). White Matter Functional Connectivity in Resting-State fMRI: Robustness, Reliability, and Relationships to Gray Matter. *Cereb Cortex* 32, 1547-1559. 10.1093/cercor/bhab181.

Wang, T., Wilkes, D.M., Li, M., Wu, X., Gore, J.C., and Ding, Z. (2020). Hemodynamic Response Function in Brain White Matter in a Resting State. *Cereb Cortex Commun* 1, tgaa056. 10.1093/texcom/tgaa056.

Zuo, X.-N., Di Martino, A., Kelly, C., Shehzad, Z.E., Gee, D.G., Klein, D.F., Castellanos, F.X., Biswal, B.B., and Milham, M.P. (2010). The oscillating brain: Complex and reliable. *NeuroImage* 49, 1432-1445. <https://doi.org/10.1016/j.neuroimage.2009.09.037>.

Reviewer #1 (Remarks to the Author):

The authors have adequately addressed my previous major concerns regarding placements of the electrode contacts and distance effects on their correlation analysis. In a second reading of the manuscript, the reviewer has identified some additional comments, basically minor, for the authors.

1. Page 3, Lines 50-54: "We further demonstrated that..., and is encoded by gene expression" sounds like the works referenced in (20-21, 25) were all undertaken by some of the authors of this manuscript, which is untrue, since it is followed immediately by "Prior studies have shown". The authors may change "We further demonstrated" to "Huang et al. further demonstrated" and add "also" right after "Prior studies".
2. Page 4, Line 57: Remove "the" (do the same to Line 25 in Page 2).
3. Page 4, Line 79: Remove "of".
4. In Supplementary Results, Line 128&188: Change "at native space" to "in native space".
5. In Supplementary Results, Figure S4-S6: Unify the ordinate scale for each of these figures to facilitate visual comparisons.

Reviewer #2 (Remarks to the Author):

The authors have fully addressed my comments. The results are relevant and represent a significant contribution to the field. The methodology and methods are significantly improved in the revised manuscript.

Reviewer #3 (Remarks to the Author):

I have previously reviewed this study and am fine with all the responses.

Reviewer #1 (Remarks to the Author):

The authors have adequately addressed my previous major concerns regarding placements of the electrode contacts and distance effects on their correlation analysis. In a second reading of the manuscript, the reviewer has identified some additional comments, basically minor, for the authors.

We thank the reviewer for the recognition of our revisions and are grateful for the additional comments.

1. Page 3, Lines 50-54: *“We further demonstrated that..., and is encoded by gene expression” sounds like the works referenced in (20-21, 25) were all undertaken by some of the authors of this manuscript, which is untrue, since it is followed immediately by “Prior studies have shown”. The authors may change “We further demonstrated” to “Huang et al. further demonstrated” and add “also” right after “Prior studies”.*

Thank you for pointing this out. We have revised accordingly (Page 3, Para 2).

Huang et al. further demonstrated that the white matter functional networks were highly reproducible across two independent datasets, and that these networks were organized into two groups with anti-correlated connectivity.

Prior studies also suggest that white matter BOLD FC could be a neuromarker for multiple psychiatric and neurological disorders, including schizophrenia, depression, Alzheimer’s disease, and Parkinson’s disease.

2. Page 4, Line 57: Remove “the” (do the same to Line 25 in Page 2).

We have deleted ‘the’ now (Page 2, Abstract; Page 4, Para 1).

Nevertheless, it remains unclear whether this white matter FC reflects underlying electrophysiological synchronization.

However, it remains unclear whether the white matter BOLD FC reflects underlying neural synchronization of intracranial electrophysiological signals in the white matter or merely a vascular phenomenon.

3. Page 4, Line 79: Remove “of”.

We have deleted ‘of’ now (Page 4, Para 3).

We tested these predictions using a multimodal dataset from a group of 16 patients with drug-resistant epilepsy, with each one completed intracranial SEEG, non-invasive resting-state BOLD fMRI, and high-quality diffusion spectrum imaging (DSI, ~24min acquisition).

4. In Supplementary Results, Line 128&188: Change “at native space” to “in native space”.

Thank you, we have changed ‘at native space’ to ‘in native space’.

Fig. S4 ... a) Evaluating the FC in native space, the Spearman’s rank correlations between the BOLD and SEEG white matter FC were similar to the main results.

Table S4 The r and p_{FDR} values for the correlations between BOLD and SEEG FC evaluated in native space.

5. In Supplementary Results, Figure S4-S6: Unify the ordinate scale for each of these figures to facilitate visual comparisons.

We appreciate your suggestion. Now we unified the ordinate scale for each of the three figures (Supplementary Information, Page 10, 11, 12). Specifically, the ordinate scale has been unified to -0.2 ~ 0.65 in Figure S4, to -0.2 ~ 0.6 in Figure S5 and Figure S6.

Reviewer #2 (Remarks to the Author):

The authors have fully addressed my comments. The results are relevant and represent a significant contribution to the field. The methodology and methods are significantly improved in the revised manuscript.

We are excited that our revisions addressed all the comments and thank the reviewer for the insightful comments again.

Reviewer #3 (Remarks to the Author):

I have previously reviewed this study and am fine with all the responses.

We thank for the reviewer’s positive feedback to the revisions.